# A generative artificial intelligence framework based on a molecular diffusion model for the design of metal-organic frameworks for carbon capture

Hyun Park[1,2,3,10], Xiaoli Yan[1,4,10], Ruijie Zhu[1,5], Eliu A. Huerta [1,6,7✉], Santanu Chaudhuri[1,4], Donny Cooper [8], Ian Foster [1,6] & Emad Tajkhorshid [2,3,9]

Metal-organic frameworks (MOFs) exhibit great promise for $CO_2$ capture. However, finding the best performing materials poses computational and experimental grand challenges in view of the vast chemical space of potential building blocks. Here, we introduce GHP-MOFassemble, a generative artificial intelligence (AI), high performance framework for the rational and accelerated design of MOFs with high $CO_2$ adsorption capacity and synthesizable linkers. GHP-MOFassemble generates novel linkers, assembled with one of three preselected metal nodes (Cu paddlewheel, Zn paddlewheel, Zn tetramer) into MOFs in a primitive cubic topology. GHP-MOFassemble screens and validates AI-generated MOFs for uniqueness, synthesizability, structural validity, uses molecular dynamics simulations to study their stability and chemical consistency, and crystal graph neural networks and Grand Canonical Monte Carlo simulations to quantify their $CO_2$ adsorption capacities. We present the top six AI-generated MOFs with $CO_2$ capacities greater than $2\,\mathrm{m\,mol\,g^{-1}}$, i.e., higher than 96.9% of structures in the hypothetical MOF dataset.

[1] Data Science and Learning Division, Argonne National Laboratory, Lemont, IL 60439, USA. [2] Theoretical and Computational Biophysics Group, NIH Resource Center for Macromolecular Modeling and Visualization, Beckman Institute for Advanced Science and Technology, University of Illinois at Urbana-Champaign, Urbana, IL 61801, USA. [3] Center for Biophysics and Quantitative Biology, University of Illinois at Urbana-Champaign, Urbana, IL 61801, USA. [4] Multiscale Materials and Manufacturing Lab, University of Illinois Chicago, Chicago, IL 60607, USA. [5] Department of Materials Science and Engineering, Northwestern University, Evanston, IL 60208, USA. [6] Department of Computer Science, University of Chicago, Chicago, IL 60637, USA. [7] Department of Physics, University of Illinois at Urbana-Champaign, Urbana, IL 61801, USA. [8] Computational Science and Engineering, Data Science and AI Department, TotalEnergies EP Research & Technology USA, LLC, Houston, TX 77002, USA. [9] Department of Biochemistry, University of Illinois at Urbana-Champaign, Urbana, IL 61801, USA. [10]These authors contributed equally: Hyun Park and Xiaoli Yan ✉email: elihu@anl.gov

Metal-organic frameworks (MOFs) have garnered much research interest in recent years due to their diverse industrial applications, including gas adsorption and storage[1], catalysis[2], and drug delivery[3]. As nanocrystalline porous materials, MOFs are modular in nature[4], with a specific MOF defined by its three types of constituent building blocks (inorganic nodes, organic nodes, and organic linkers) and topology (the relative positions and orientations of building blocks). MOFs with different properties can be produced by varying these building blocks and their spatial arrangements. Nodes and linkers differ in terms of their numbers of connection points: inorganic nodes are typically metal oxide complexes whereas organic nodes are molecules, both with three or more connection points; organic linkers are molecules with only two connection points.

MOFs have been shown to exhibit superior chemical and physical $CO_2$ adsorption properties. In appropriate operating environments, they can be recycled, for varying numbers of times, before undergoing significant structural degradation. However, their industrial applications have not yet reached full potential due to stability issues, such as poor long-term recyclability[5], and high moisture sensitivity[6]. For instance, the presence of moisture in adsorption gas impairs a MOF's $CO_2$ capture performance, which may be attributed to MOFs having stronger affinity toward water molecules than to $CO_2$ molecules[7]. Other investigations of MOF stability issues[8–12] have shown that the gas adsorption properties of MOFs can be enhanced by tuning the building blocks used. Yet progress has been difficult due to the enormous chemical space of building blocks, which makes exhaustive search with traditional experimental methods impractical[13].

**Related work**. Previous efforts in the search for new MOF structures with exceptional gas adsorption properties include:

Database search methods. These methods apply filters to identify optimal candidates in large databases of MOF structures plus calculated properties obtained with molecular simulations. For example, Ref. [14] shows how to search for MOFs with high $CO_2$ uptake when moisture is present using the experimental MOF database (CoRE DB), while Ref. [15] shows how to search for MOFs with high $CH_4/H_2$ selectivity in both CoRE DB and the Cambridge Structural Database non-disordered MOF subset.

Machine Learning (ML)-assisted screening. Building large MOF databases require expensive molecular simulation calculations for every target structure. ML-assisted screening avoids this difficulty by using a regression model trained on a smaller training set to predict target properties of a large number of new test structures. This approach has been widely applied to gas adsorption and separation property predictions of MOFs, including $CO_2/H_2$ separation[16] and $CH_4$ adsorption[17]. ML-assisted methods use of geometrical features (e.g., dominant pore size, void fraction[18]) and chemical features (e.g., atom type, electronegativity[19]). A compound feature, the atomic property weighted radial distribution function[20] has been shown to improve regression model performance in finding MOF structures with high $CO_2$ capacity[21]. This feature is constructed by using both local geometry and chemistry of atomic sites in MOF structures. As an alternative to the use of hand-engineered ML features, neural networks have been applied to MOF research. In Ref. [22], the Atomistic Line Graph Neural Network, trained on the hMOF dataset, was applied to search CoRE DB for high-performing MOFs for carbon capture[23].

Generative modeling. Candidates are not drawn from a database but are generated de novo via methods that produce novel compounds that have a desired set of chemical features. For instance, the Supramolecular Variational Autoencoder[24] uses a semantically constrained graph-based canonical code to encode MOF building blocks. The variational autoencoder framework structure allows this model to interpolate between existing MOF structures and enables isoreticular optimization of MOF structures toward higher $CO_2$ capacity and $CO_2/N_2$ selectivity.

This work. Proposed generative models include variational autoencoders, generative adversarial networks, normalizing flows, autoregressive models, and diffusion models[25]. Here, we adopt a diffusion model named DiffLinker to generate novel MOF linkers. Diffusion models use a probability distribution and Markovian properties to generate new data via forward diffusion and backward denoising steps. First, Gaussian noise is added to the input samples to yield noisy data. A neural network is then trained to learn what noise was added. The trained network is then used to reversely transform (i.e., denoise) the noisy data back into target samples that resemble molecules from the training data distribution. This method has been widely applied to drug discovery to speed up the design of new ligands and ligand-protein complexes[26–29]. Two broad categories of molecular representation schemes have been used in diffusion model-based ligand generators: molecular graphs and 3D coordinates. In the former case (Digress/Congress[30] is an example), atoms are represented as nodes and bonds as edges. In the latter case, the 3D atomic coordinates of molecules are generated directly, as in DiffLinker[31] and E(3) equivariant diffusion model (EDM)[32]. Given the success of diffusion models in drug discovery[26,28,29,33,34], we demonstrate how to transfer the idea to the iso-reticular design of MOF structures by varying MOF linkers while fixing node and topology. The diffusion model is used specifically for MOF linker design.

Our proposed approach, GHP-MOFassemble, is a novel high-throughput computational framework to accelerate the discovery of MOF structures with high $CO_2$ capacities and synthesizable linkers. While previous attempts to discover useful MOFs via computational methods have proceeded via high-throughput screening of existing datasets[14,35,36], an approach that necessarily limits the search to known MOFs. In contrast, GHP-MOFassemble probes the MOF design space by employing a molecular generative diffusion model, DiffLinker[31], to generate chemically diverse and unique MOF linkers from a set of molecular fragments, which it assembles with pre-selected metal nodes to form novel MOF structures. It then screens those structures with a pre-trained regression model, a modified version of Crystal Graph Convolutional Neural Network (CGCNN)[37], to identify high-performing MOF candidates for carbon capture. In addition, we apply molecular dynamics (MD) and grand canonical Monte Carlo (GCMC) to further down-select stable AI-generated MOFs and compute more credible $CO_2$ adsorption capabilities. We demonstrate the utility of our framework by applying it to the rational design of MOFs with pcu topology and three types of inorganic nodes: Cu paddlewheel, Zn paddlewheel, and Zn tetramer.

## Results

Here we describe the key components of GHP-MOFassemble, and present a new set of AI-generated MOF structures with high $CO_2$ capacities. A detailed description of the approaches used to obtain these findings is provided in Methods.

**Analysis of the hMOF dataset**. The three most frequent node-topology pairs in the hMOF dataset, accounting for around 74% of its MOFs, are the Cu paddlewheel-pcu, Zn paddlewheel-pcu, and Zn tetramer-pcu, which amount to 102,117 hMOF structures (29,714 + 28,529 + 43,874 from first column of Table 1). Out of these 102,117 structures, we only used those with correctly parsed

**Table 1 Most frequent node-topology pairs in the hMOF dataset.**

| node-topology | total MOFs | HP-MOFs | HP-MOFs with three linkers | unique linker SMILES | unique molecular fragment conformers |
|---|---|---|---|---|---|
| Cu PW-pcu | 29,714 | 1458 | 1016 | 3330 | 180 |
| Zn PW-pcu | 28,529 | 1314 | 834 | 3221 | 162 |
| Zn TM-pcu | 43,874 | 2129 | 1388 | 3265 | 198 |

Properties of the hMOF dataset. PW, TM, and HP-MOF stand for paddlewheel, tetramer, and high-performing MOF, respectively.

**Table 2 Number of linkers at each step of the MOF assembly process.**

| Method | Total | Cu PW-pcu | Zn PW-pcu | Zn TM-pcu |
|---|---|---|---|---|
| DiffLinker | 64,800 | 21,600 | 19,440 | 23,760 |
| Hydrogen addition | 56,257 | 18,979 | 17,126 | 20,152 |
| Dummy atom identification | 16,162 | 4964 | 4450 | 6748 |
| Element filter | 12,305 | 3702 | 3441 | 5162 |

Statistical summary of the number of linkers after each step categorized by the corresponding MOF types. PW and TM stand for paddlewheel and tetramer, respectively. Element filter means that linkers with S, Br and I elements are removed.

MOFids and valid Simplified Molecular-Input Line-Entry System (SMILES), i.e., 78,238 MOFs.

**Linker generation and evaluation**. For all three MOF types, we start with 540 (i.e., = 180 + 162 + 198 from Table 1 last column) unique molecular fragments extracted from high-performing hMOF structures, and use DiffLinker to generate new MOF linkers, with the number of sampled connection atoms ranging from five to 10 (total six connection atom types). For each linker, sampling is performed 20 times. Therefore, the resulting candidate pool contains a total of 64,800 linkers, i.e., 540 fragments × 20 independent sampling steps × six unique number of connection atoms. Since these linkers only contain heavy atoms, to generate all-atom linker molecules, we apply openbabel to add hydrogen atoms, which results in 56,257 linkers after removing linkers with erroneous hydrogen assignments. Next, dummy atom identification is performed to generate information that enables assembly with metal nodes. A total of 16,162 linkers with dummy atoms are generated. These linkers are then passed through the element filter, which removes linkers that contain S, Br, and I, further reducing the number to 12,305 linkers. The number of linkers in each step are summarized in the Total column of Table 2.

**MOF assembly**. We generate new MOF structures by assembling three randomly selected DiffLinker-generated linkers with one of the three most frequent nodes in the hMOF dataset. Random sampling of 12,305 linkers ensures that the selected linkers cover a large design space. For each node-linker-topology combination, we considered four levels of catenation (cat0, cat1, cat2, cat3). We generated a total of 120,000 MOFs as follows: we have four catenation levels and three node candidates. The random sampling of 10,000 linkers for each catenation level-node candidate pair generates 120,000 total MOFs of different catenation-node-linker combinations.

**MOF geometry examination**

*Inter-atomic distance check*. We used Pymatgen to read each MOF structure's CIF file and to extract the pairwise distance matrix. Diagonal entries of the distance matrix are discarded as they signify the distance between an atom and itself. The off-diagonal values are tested for minimum inter-atomic distance threshold. The inter-atomic distance threshold is predetermined by an experimental database, OChemDb[38]. If a MOF has at least one entry of lower than threshold inter-atomic distance, the MOF is discarded. 78,796 of the 120,000 assembled MOFs passed this test.

*Pre-simulation check*. We used the open source library, cif2lammps[39], to automatically assign the Universal Force Field for Metal-Organic Frameworks (UFF4MOF)[40,41] to MOFs and generate LAMMPS[42] input files. This step ensures that all atomic structures and bonding appearing in each MOF structure are chemically valid within the scope of UFF4MOF. For example, if a Zn atom is bonded to a C atom, or if an O atom is bonded to four other atoms, etc., these structures will be identified as invalid and will be discarded. Of the 78,796 MOFs that passed geometry and inter-atomic distance checks, 18,770 passed this pre-simulation check, and thus had LAMMPS input files generated.

**Regression model for MOF $CO_2$ capacity prediction**. To reduce the number of LAMMPS simulations, a screening step of the MOFs' adsorption performance is conducted on the 18,770 MOFs described above using a modified version of the CGCNN model that we introduced in Ref. [37]. Here, we used MOF structures from the hMOF dataset as well as their $CO_2$ capacities at 0.1 bar as input data. We split the hMOF dataset into three independent sets: 80% for training, 10% for validation, and 10% for testing. Using this data split, we trained three CGCNN models using random initialization of weights for each of them. When we use this model ensemble to infer the $CO_2$ capacities of newly AI-generated MOF structures, we take the average of the predictions made by the three independent models as the predicted $CO_2$ adsorption capacity. Out of the 18,770 AI-generated MOF structures, a total of 364 were predicted to be high-performing MOFs with $CO_2$ capacity higher than $2 \text{ m mol g}^{-1}$ at 0.1 bar. Specifically, we used ensemble model prediction mean plus standard deviation as $CO_2$ adsorption capacity value to be higher than $2 \text{ m mol g}^{-1}$ as the threshold. The standard deviation consideration is to assure we account for statistical errors of adsorption predictions by our ensemble model. Categorization of the predicted high-performing MOFs by node-topology pair and catenation level indicates that for all three node types most of the high-performing MOFs are cat2 and cat3.

**Structural validation of AI-generated MOFs with molecular dynamics simulations**. We now examine the stability and porous properties of the 364 AI-generated MOFs described in the previous section using MD simulations with LAMMPS[42]. For each MOF, a $2 \times 2 \times 2$ supercell structure is equilibrated under a tri-clinic isothermal-isobaric ensemble (i.e., NPT) at $\langle p \rangle = 1$ atm and $\langle T \rangle = 300K$, such that the cell lengths and angles of the MOF structure can be equilibrated. The NPT simulations are run for

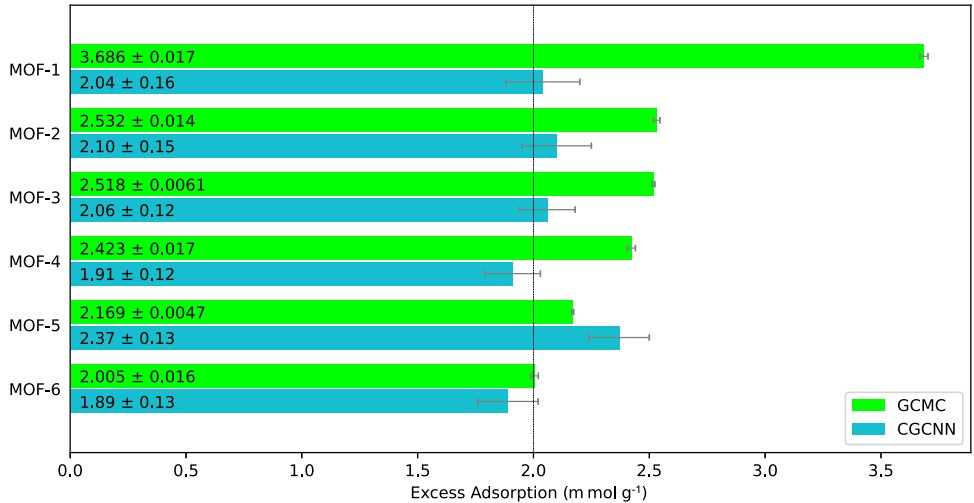

**Fig. 1 CO$_2$ adsorption values at 0.1 bar and 300K.** CO$_2$ adsorption capacity values at a pressure and temperature of 0.1 bar and 300 K, respectively, for the top six AI-generated MOFs' according to grand canonical Monte Carlo (GCMC) simulations and our modified crystal graph convolutional neural network (CGCNN) model.

400,000 steps with step size of 0.5 femtoseconds. The relative changes of the lattice parameters before and after the simulation are inspected to evaluate how well the MOF structure is maintained throughout the equilibrium MD simulation. The MOFs with higher than 5% changes in any of the lattice parameters ($a$, $b$, $c$, $\alpha$, $\beta$, $\gamma$) are discarded (first three are lattice vector lengths and the latter three are lattice angles) between the initial state and the equilibrated state. The MOFs with higher than 5% changes in any of the lattice parameters ($a$, $b$, $c$, $\alpha$, $\beta$, $\gamma$) are discarded. This process produces 102 MOFs that satisfied the previously defined criteria, i.e., < 5% in lattice parameter change[40].

**Property validation of AI-generated MOFs with grand canonical Monte Carlo (GCMC) simulations.** These 102 MOFs are further examined with GCMC simulations to calculate their CO$_2$ adsorption capacity at a pressure and temperature of 0.1 bar and 300 K, respectively. RASPA[43] is used to conduct these GCMC CO$_2$ adsorption simulations. As demonstrated in Fig. 1, 6 MOFs candidates were found to have CO$_2$ adsorption capacity higher than 2 m mol g$^{-1}$ by the GCMC simulations. The 3D structure of these six high performing, AI-generated MOFs is presented in Fig. 2.

## Discussion

GHP-MOFassemble combines novel generative and graph AI applications, as well as a comprehensive screening workflow that combines various modeling methods with increasing levels of chemistry awareness and increasing levels of computational cost. When we deployed and optimized GHP-MOFassemble on the Delta supercomputer, unless indicated otherwise, we found that:

- We AI-assembled 120,000 MOFs within 33 minutes using multiprocessing on 28 cores in the ThetaKNL supercomputer at the Argonne Leadership Computing Facility (ALCF).
- We screened these 120,000 MOFs in 40 minutes using multiprocessing on 128 cores, and identified 78,796 MOFs with valid bond lengths.
- We further screened these 78,796 MOFs and identified 18,770 MOFs with valid chemistry, using UFF4MOF as a reference, within 205 minutes using multiprocessing on 128 cores.
- We then used our AI ensemble of CGCNN models to estimate the CO$_2$ capacity of these 18,770 MOFs, and

identified 364 high-performing, AI-generated MOFs with CO$_2$ capacity higher than 2 m mol g$^{-1}$ at 0.1 bar. AI inference was completed in 50 minutes using one NVIDIA A40 GPU.

In brief, from assembly to selection of high-performing MOFs, GHP-MOFassemble completes the analysis within 5 hours and 7 minutes. Once we have selected 364 AI-generated, high-performing MOFs, we carry out the most compute-intensive part of the analysis, namely:

- We used the LAMMPS code to equilibrate 364 MOFs with 200-picosecond NPT MD simulations with the UFF4MOF force field. Each of these simulations is completed within 11 minutes, on average, using between 6 to 14 MPI processes, depending on the number of atoms in the MOF. We identified 102 stable MOFs whose lattice parameters changed less than 5% throughout the MD simulations.
- Finally, each of these 102 MOFs were further examined with GCMC simulations, and a total of six MOF candidates were found to have CO$_2$ capacity higher than 2 m mol g$^{-1}$, which corresponds to the top 5% of the hMOF dataset. Each of these 102 GCMC simulations is completed within six hours, on average, using one CPU core.

Recent studies[44–47] indicate that several functional groups play an important role in determining MOF CO$_2$ capacity, including carboxylic acid (-COOH), primary amine (-NH$_2$), hydroxyl (-OH), and nitrile (-CN). These functional groups affect MOF CO$_2$ capacity due to their interaction with CO$_2$ molecules through charge redistribution[46], hydrogen bond interaction[45], and electrostatic interaction[45]. Thus, we analyzed the functional groups of linkers in the 364 AI-generated high-performing MOF structures and compared them with high-performing hMOF structures. The results of this analysis are presented in Fig. 3. We observe that linkers in high-performing hMOF structures have a higher proportion of carboxylic groups, whereas hydroxyl groups appear more often in the predicted high-performing MOFs generated by our framework. Moreover, the percentages of primary amine and nitrile in linkers are similar in both cases. Note that some substructures appear more often than others. For example, ring structures appear in most of the linkers in the top candidates, and many of the DiffLinker-generated molecular substructures (in between the terminal fragments) also contain

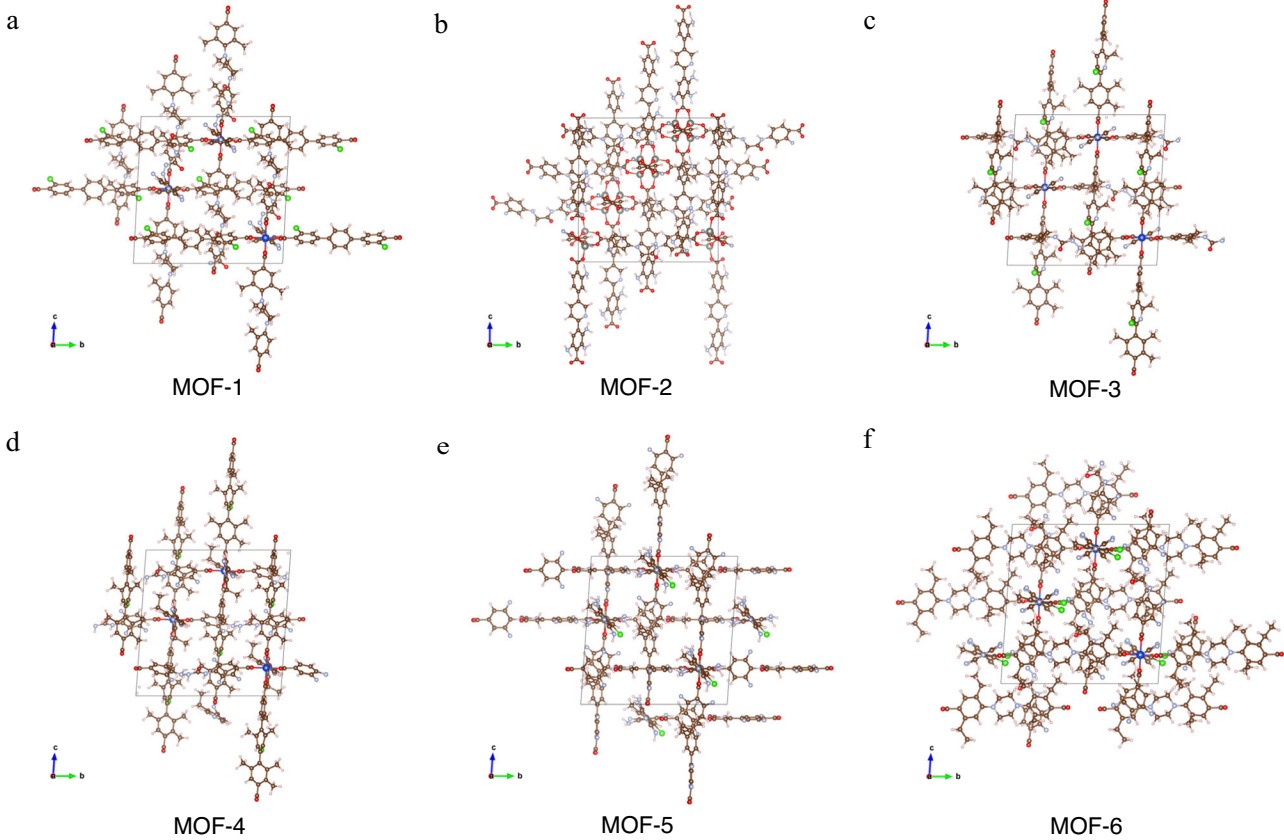

**Fig. 2 Visualization of the crystal structure of AI-generated MOFs. a–f** Crystal structure of the top six AI-generated MOFs. The color code used to represent atoms is: carbon in grey, nitrogen in dark blue, fluorine in cyan, zinc in purple, hydrogen in white, and lithium in green.

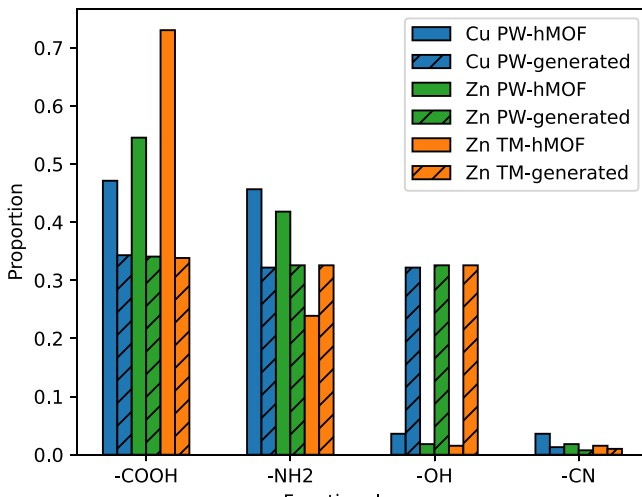

**Fig. 3 Functional groups in high-performing MOFs.** Comparison of the proportion of selected functional group that appear in high-performing AI-generated MOFs and MOFs in the hMOF dataset.

ring structures. The frequent occurrence of rings in linkers of high-performing MOFs may be due to the presence of $CO_2$-ring interaction, as reported in Ref. [48]. In addition, rings in the linkers may increase MOFs' structural rigidity due to less rotatable bonds.

In summary, the entire discovery process from generative AI MOF assembly to detailed GCMC simulations of top AI-

predicted MOFs can be completed within 12 hours using distributed computing in modern supercomputing platforms. The entire workflow may be further accelerated by scaling up any of the subprocesses using more cores or by distributing AI inference over more GPUs. Furthermore, the various pieces of GHP-MOFassemble may be assembled to create a standalone workflow that combines MD, density functional theory, and GCMC simulations to expose our generative AI model to a larger set of chemically diverse, high performing MOFs. Through online learning methods one may continually guide generative AI until it consistently assembles high performing MOFs. A study of this nature will be pursued in future work.

## Methods
Our GHP-MOFassemble framework has three components:

- **Decompose**. We use a molecular fragmentation algorithm to decompose the MOF linkers found in high-performing MOF structures within the hMOF dataset[22]—an open source dataset that provides, for each of 137,652 hypothetical MOF structures, and corresponding MOFid, MOFkey, geometric features, and isotherm data for six adsorption gases ($CO_2$, $N_2$, $CH_4$, $H_2$, Kr, Xe) at 0.01, 0.05, 0.1, 0.5, and 2.5 bar—covering the pressure ranges of cyclic adsorption gas separation processes in industrial applications. The adsorption properties of gases provided in this dataset were calculated using GCMC calculations[22], as described in Ref. [49].

- **Generate**. We use a diffusion model to generate new MOF linkers. We then screen the AI-generated linkers by removing linkers with S, Br and I elements (we call this step "element filter"), and evaluate their quality using five

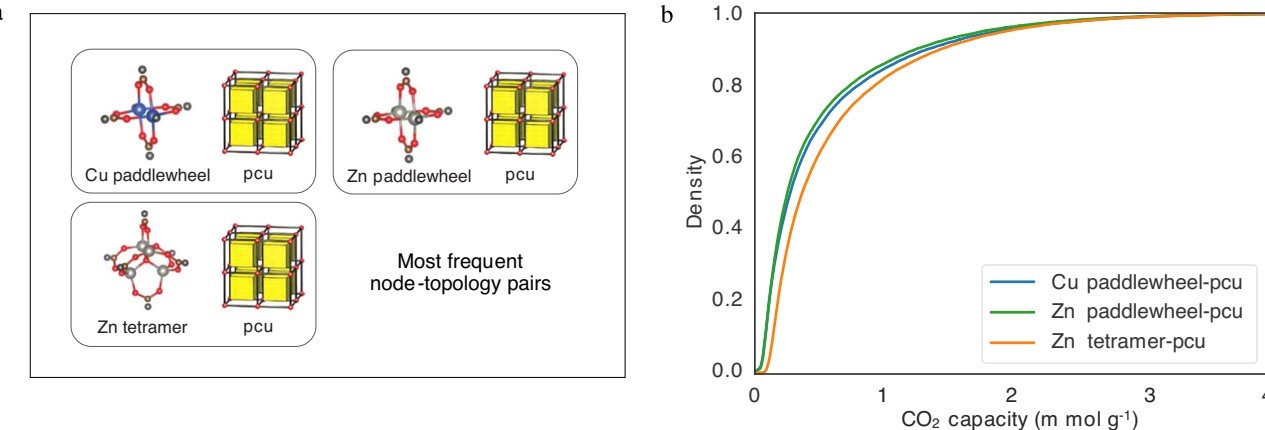

**Fig. 4 Properties of the hMOF dataset. a** Depictions of the most frequent node-topology pairs in hMOF structures. **b** Cumulative distribution functions of their 0.1 bar $CO_2$ capacities.

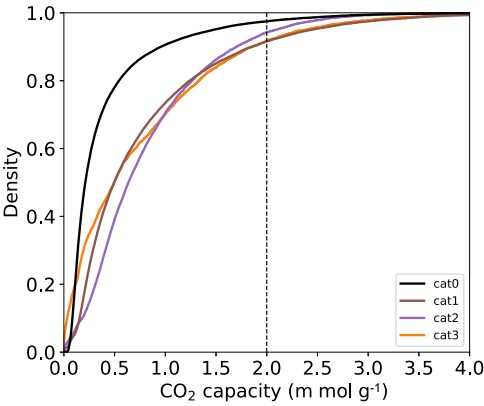

**Fig. 5 $CO_2$ capacities of hMOF structures at different catenation levels.** Empirical cumulative distribution functions of MOFs in the hMOF dataset at 0.1 bar at different catenation levels. The x axis is capped at 4 m mol g$^{-1}$ to preserve details.

different scores to quantify their synthesizability (SAScore and SCScore), validity, uniqueness, and internal diversity. Linkers that passed the element filter are then assembled with one of three pre-selected nodes into new MOF structures in the pcu topology.

- **Screen and Predict**. We check for inter-atomic distances with pre-simulation checks to ensure we filter poor AI-generated MOF structures from previous steps. We then use a pre-trained regression model to predict the $CO_2$ capacities of the newly generated MOF structures. Lastly, we perform MD and GCMC simulations to obtain stable and high-performing MOF structures.

In the following sections we describe the datasets we have used, and each of the GHP-MOFassemble components in detail.

**The hMOF dataset**. We selected the most common topologies of the hMOF dataset for this work, i.e., Cu paddlewheel-pcu, Zn paddlewheel-pcu, and Zn tetramer-pcu, i.e., 78,238 MOFs with correctly parsed MOFids and valid SMILES. We gather the numbers of molecular fragments in Table 1, produced through the *Fragment* step of our GHP-MOFassemble framework, described below. In Table 1 high-performing MOFs (second column) with three parsed linkers are selected (third column), and their unique linkers are parsed by using the MMPA (Matched Molecular Pairs Algorithm) algorithm. The cumulative

distribution functions of the $CO_2$ capacities of these three types of MOFs are shown in the right panel of Fig. 4.

In Table 1, the number of output molecular fragment conformers (last column) is much less than the number of unique linker SMILES (second to last column) for two reasons. First, around 56% of linkers did not pass the valency check, which may be due to the intrinsic limitations in the parsing of SMILES of MOF linkers. Second, around 90% of linkers that passed the valency check share similar molecular fragments, and thus many duplicated molecular fragments exist. The successfully parsed unique molecular fragment conformers are subsequently used for linker generation.

Figure 5 shows the empirical cumulative distribution functions of the $CO_2$ capacities of hMOF structures at different catenation levels (i.e., MOFs with interpenetrated lattices). Therein, we observe that a higher percentage of catenated MOFs (cat1, cat2 and cat3) are high-performing, as compared to the uncatenated MOFs (cat0). This result confirms that catenation is an important factor when designing new MOF structures with high $CO_2$ capacity. This observation is consistent with other studies in the literature[50,51], which indicate that even though catenation reduces pore size and surface areas, catenated MOFs generally have higher $CO_2/H_2$ selectivities because MOF-$CO_2$ interactions are enhanced as a result of the strong confinement of $CO_2$ with a much lower adsorption surface. Thus, the results presented in Fig. 5 using the hMOF dataset indicate that $CO_2$ working capacities of catenated MOFs are higher than their noncatenated counterparts. As we discuss below, we found a similar pattern in AI-generated MOFs. Catenated MOFs were generated by using site translation method, which is achieved by displacing the reference lattice along the diagonal line of the unit cell. For all four catenation levels, the amount of relative lattice displacement is given in Table 3. The numbers are fractional displacements relative to the unit cell diagonal.

Figure 6 shows the pairwise relationships of the $CO_2$ capacities of hMOF structures at five pressures. We observe a strong correlation of $CO_2$ capacities between 0.05 bar and 0.1 bar, with a Pearson's correlation coefficient of 0.86. For other pairs of pressures, the $CO_2$ capacities are only weakly correlated, with a decreasing correlation for larger pressure differences. Moreover, the distributions of $CO_2$ capacities at all pressures exhibit long tails at high value ranges, which indicate that the majority of MOFs are low-performing and high-performing MOFs are uncommon.

**Decomposing MOF linkers into molecular fragments**. The first component of the GHP-MOFassemble framework, **Decompose**, decomposes linkers from high-performing MOF structures in the

hMOF dataset into their molecular fragments. As illustrated in Fig. 7, this process applies the following three steps to a specified node-topology pair, which in the figure is the most frequent node-topology pair in the hMOF dataset, Zn tetramer-pcu. Note that for the pcu topology, one linker is orientated along each of the x, y, and z directions, and thus at most three unique linker types are possible.

1. *Select*: We select high-performing MOFs with the given node-topology pair from hMOF. We define here a high-performing MOF structure as one with $CO_2$ capacity higher than 2 m mol g$^{-1}$ at 0.1 bar, which corresponds to the top 5% of $CO_2$ capacity of the hMOF dataset.
2. *Extract*: We extract the linker Simplified Molecular-Input Line-Entry System (SMILES) strings for each high-performing MOF identified in *Select* step, eliminate those with more than three unique linker types, and assemble the remaining into a linker dataset.
3. *Fragment*: We fragment the linkers produced in *Extract* step to obtain their chemically relevant fragment-connection atom pairs, which we assemble into a molecular fragment dataset.

The extraction of linkers in *Extract* step is straightforward because the MOFid of each hMOF structure specifies the SMILES strings of its constituent metal nodes, linkers, as well as a format

signature and a topology code[52]. Together, these elements uniquely define the topology and building blocks of a given MOF structure.

*Fragment* step use MMPA[53] as implemented in DeLinker[54] to generate molecular fragments of a given molecule by breaking chemical bonds between atom pairs. We set the minimum number of connection atoms, the minimum fragment size, and the minimum path length to 3, 5, and 2, respectively. Moreover, we only consider the case where the fragments are at least two atoms away from each other. The chemically relevant fragment-connection atoms pairs are then used to form the molecular fragment dataset.

**Generating new MOF structures**. The **Generate** component employs the pre-trained diffusion model DiffLinker[31] to generate new MOF linkers, and then assembles those new linkers with one of three pre-selected nodes into new MOF structures in the pcu topology. It comprises three steps: *Diffuse and Denoise*; *Screen and Evaluate*; and *Assemble*. An example of these three steps for Zn tetramer-pcu MOFs is shown in Fig. 8.

*Diffuse and Denoise*. We apply the pre-trained diffusion model DiffLinker[31] to generate new linkers based on the molecular fragments outputted by the **Decompose** component. This model connects the molecular fragments supplied as input (also known as context) with a specified number of sampled atoms. With less sampled atoms, straight chains or branched chains may be obtained, while with more sampled atoms, ring structures may be present. Moreover, during the denoising process, the species of the sampled atoms may change. In this work, we vary the number of sampled atoms from 5 to 10 to ensure that the generated linkers have a diverse set of chemistry and substructures.

DiffLinker applies a denoising process to determine the atomic species and Cartesian coordinates of the sampled atoms by using a decoder neural network architecture named E(3)-Equivariant Graph Neural Network (EGNN)[55]. This graph neural network

| Table 3 Relative lattice displacement at each catenation level. | | | | |
|---|---|---|---|---|
| catenation | lattice1 | lattice2 | lattice3 | lattice4 |
| cat0 | (0,0,0) | / | / | / |
| cat1 | (0,0,0) | $(\frac{1}{2},\frac{1}{2},\frac{1}{2})$ | / | / |
| cat2 | (0,0,0) | $(\frac{1}{3},\frac{1}{3},\frac{1}{3})$ | $(\frac{2}{3},\frac{2}{3},\frac{2}{3})$ | / |
| cat3 | (0,0,0) | $(\frac{1}{4},\frac{1}{4},\frac{1}{4})$ | $(\frac{1}{2},\frac{1}{2},\frac{1}{2})$ | $(\frac{3}{4},\frac{3}{4},\frac{3}{4})$ |
| Amount of lattice displacement for catenated MOFs. | | | | |

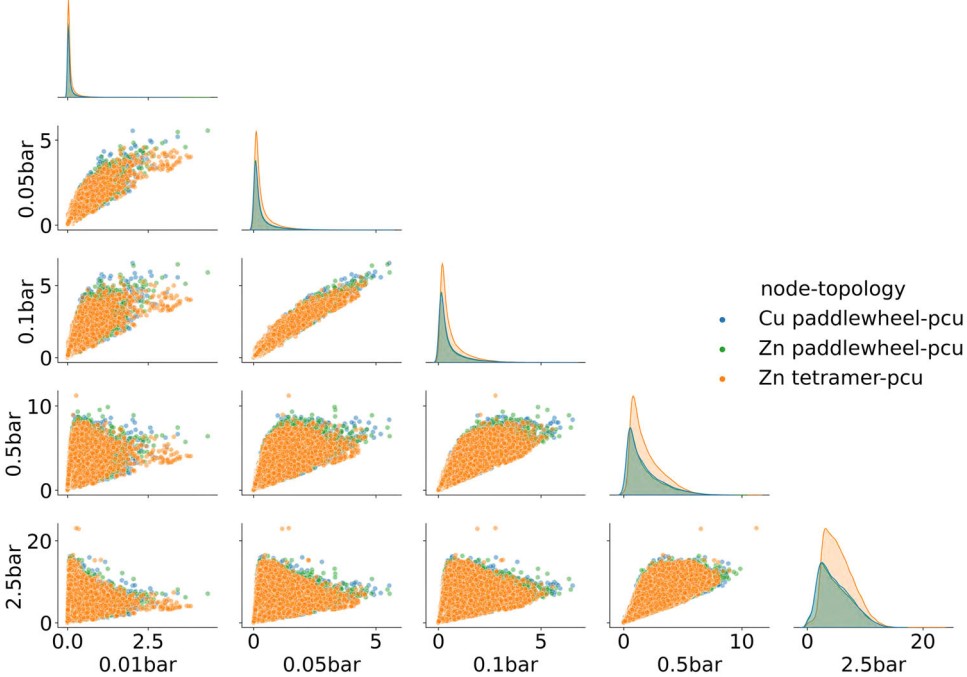

**Fig. 6 CO$_2$ capacities of hMOF structures at different adsorption pressures.** Pairplot of $CO_2$ capacities of hMOF structures. The $(x, y)$ axes represent adsorption pressures. Different colors indicate MOFs with different node-topology pairs.

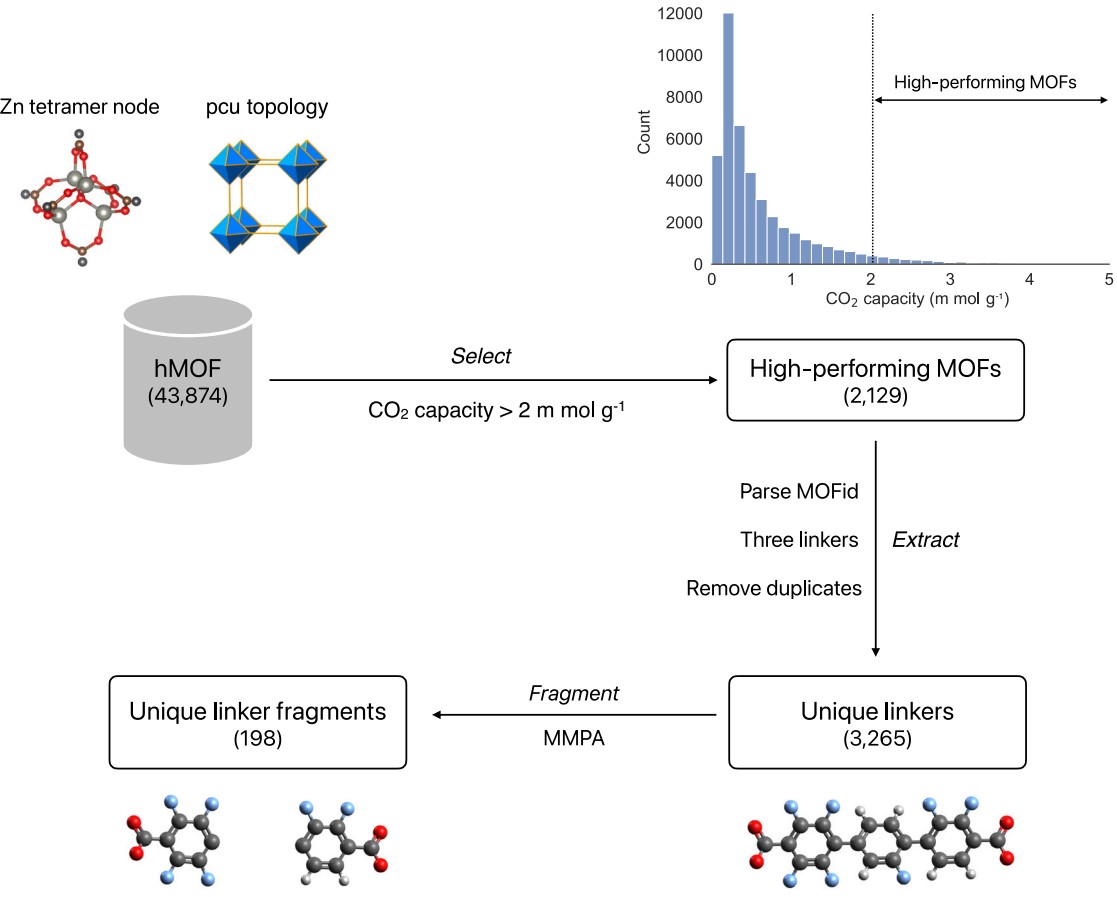

**Fig. 7 First component of the GHP-MOFassemble framework.** Schematic representation of the **Decompose** component, which consists of three steps, which we showcase for Zn tetramer-pcu MOFs. First, the *Select* step selects the Zn tetramer-pcu MOFs in high-performing hMOF structures. Next, the *Extract* step extracts unique linkers from those MOFs. Finally, the *Fragment* step generates unique linker fragments. The color scheme of elements is: carbon in grey, oxygen in red, nitrogen in blue, and hydrogen in white. See Table 1 for unique linker fragments of Zn tetramer node.

predicts zero-mean centered Cartesian coordinates noise and one-hot encoding noise of atomic species, and accounts for equivariance due to translation, rotation, and reflection of molecules, i.e., Euclidean Group 3 (E(3)). In this work, we use the pre-trained DiffLinker model that was trained on the GEOM dataset[56], which contains 37 million molecular conformations for over 450,000 molecules. Such diverse training data enables DiffLinker to sample linker molecules with high chemical and structural diversity.

The outputs of DiffLinker are the 3D coordinates and the atomic species of heavy atoms in the linker molecules, rather than SMILES strings or 2D graphs. Since hydrogen atoms are implicit in the DiffLinker model, their 3D coordinates are not outputted by the model. To generate the spatial configurations of all-atom molecules, we employ openbabel to convert the DiffLinker generated molecules to SMILES strings, which in turn are used to generate 3D configurations of linkers with explicit hydrogen atoms. Openbabel uses distance-based heuristics to determine bond connectivity in a given molecule[57], which is critical to the assignment of the number of hydrogen atoms. Finally, after the hydrogen addition step, we identify the dummy atoms which contains information about how the linkers connect with metal nodes.

The diffusion process involves two consecutive steps. The first step adds Gaussian noise to the original data (i.e., $\mathbf{x}$), yielding noisy input (i.e., $z_t$ at time $t$). Next, the denoising step applies neural network-based noise removal operation. Mathematically,

the following Markovian properties and equations are satisfied:

$$q(z_t|z_{t-1}) = \mathcal{N}(z_t; \bar{\alpha}_t z_{t-1}, \bar{\sigma}_t^2 \mathbf{I}), \tag{1}$$

$$q(z_t|\mathbf{x}) = \mathcal{N}(z_t; \alpha_t \mathbf{x}, \sigma_t^2 \mathbf{I}), \tag{2}$$

$$p(z_{t-1}|z_t) = q(z_{t-1}|\mathbf{x}, z_t), \tag{3}$$

$$q(z_{t-1}|\mathbf{x}, z_t) = \mathcal{N}(z_{t-1}; \mu_t^\theta(\mathbf{x}, z_t; \alpha_t, \sigma_t), \xi_t \mathbf{I}), \tag{4}$$

$$\mu_t^\theta(\mathbf{x}, z_t; \alpha_t, \sigma_t) = A_t z_t + B_t \mathbf{x}, \tag{5}$$

$$\hat{\mathbf{x}} = \mathbf{x} = C_t z_t - D_t \epsilon_t^\theta(z_t, t), \tag{6}$$

where $q$ and $p$ are the probability density functions during forward (diffusion) and reverse (denoising) process, respectively. $\mathcal{N}$ stands for normal distribution. $\bar{\alpha}_t = \alpha_t / \alpha_{t-1}$, and $\bar{\sigma}_t^2 = \sigma_t^2 - \bar{\alpha}_t^2 \sigma_{t-1}^2$. $\mathbf{I}$ is an identity matrix that computes an isotropic Gaussian upon multiplying with a constant (e.g., $\sigma_t^2$); $\alpha_t$ and $\bar{\alpha}_t$ are signal controls, i.e., meaningful information during training and generative steps; and $\sigma_t$ and $\bar{\sigma}_t$ are noise controls, i.e., noisy information used for diversity during training and generative steps. The subscript $t = 0, 1,...,T$ is the time step at which a molecule is generated (a.k.a. denoised or diffused). $\xi_t, A_t, B_t, C_t$, and $D_t$ are constants consisting of $\alpha_t, \bar{\alpha}_t, \sigma_t$, and $\bar{\sigma}_t$[31].

Equations (1) and (2) represent a diffusion process that is conditioned on the previous value, $z_{t-1}$, and initial value, $\mathbf{x}$, to predict a current value, $z_t$. The denoising processes are described

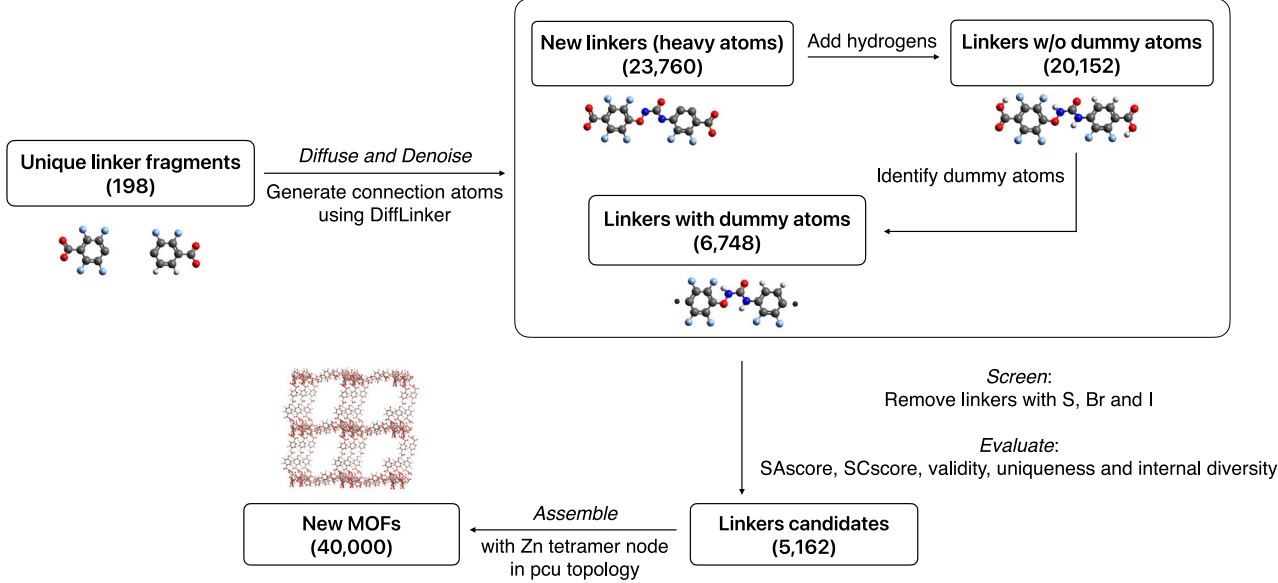

**Fig. 8 Second component of the GHP-MOFassemble framework.** The **Generate** component of the GHP-MOFassemble framework involves *Diffuse and Denoise, Screen and Evaluate,* and *Assemble* steps, as shown here for Zn tetramer-pcu MOFs (see Table 1 for unique linker fragments of Zn tetramer node). First, we generate new linkers via the *Diffuse and Denoise* step. We then add hydrogen atoms to ensure correct valency of the generated linkers. Next, we identify dummy atoms by replacing the two carbon atoms in the carboxyl groups of each generated linker with dummy atoms. Linkers with dummy atoms then undergo the *Screen and Evaluate* step, where we remove those with S, Br, and I elements, since these three elements are present in the GEOM dataset but not in the hMOF dataset. As a result, this step reduces the number of potential linkers to 5162. The generated linkers' molecular statistics are quantified using five metrics, including SAscore, SCscore, validity, uniqueness, and internal diversity. Finally, in the *Assemble* step, we build 40,000 new MOFs with Zn tetramer node in pcu topology. As before, the color scheme of elements is: carbon in grey, oxygen in red, nitrogen in blue, and hydrogen in white. See last column of Table 2 regarding this component.

by Equations (3) and (4), which predict a previous value conditioned on (either real or predicted) initial value and current value. Equations (5) and (6) provide a detailed evolution for $\mu_t^\theta$ which represents the learned denoising mean parameterized by $\theta$, as well as **x**, a real initial value, approximated by $\hat{\mathbf{x}}$. The training objective is to learn a neural network, i.e., EGNN, to predict a denoising value so that random noise can be denoised to a physically and chemically valid molecule during the generative process. In practice, we predict $\epsilon_t^\theta$ (learned noise value) rather than $\hat{\mathbf{x}}$ directly (i.e., Equations (5) and (6)) for better prediction[58]. Since DiffLinker generates chemically valid molecules, EGNN needs to take more regularization into account, such as E(3) and O(3) (i.e., orthogonal group in dimension 3: rotations and reflections). Thus, invariant features such as one-hot and equivariant features such as atomic coordinates updates need to be considered[31].

*Screen and Evaluate.* To ensure consistency of elements in hMOF linkers and AI-generated MOF linkers, we manually filter out generated linkers that contain elements not present in the hMOF dataset. Since the pre-trained DiffLinker model we used in this work was trained on the GEOM dataset[56], a total of nine heavy elements may be sampled, including C, N, O, F, P, S, Cl, Br, and I. Among these elements, S, Br, and I do not appear in the hMOF dataset, therefore we remove generated linkers that contain these three elements. For each molecular fragment, we then perform sampling 20 times. Each time the model samples a different molecule may be obtained because of the random and probabilistic nature of denoising process. The probabilistic nature of diffusion model enables it to generate linkers from an extensive linker design space beyond that of the hMOF dataset.

We then use five metrics to evaluate the quality of the remaining linkers. The first two metrics are commonly used heuristic measures of synthesizability: the synthetic accessibility score (SAscore), and the synthetic complexity score (SCscore)[59].

The SAscore, as defined by Ertl and Schuffenhauer[60], is based on analysis of one million PubChem molecules, and combines fragment contributions from molecule substructures with a complexity penalty that accounts for molecular size and for structural features of molecules such as the presence of rings. The SCscore is computed by using a neural network trained on 12 million chemical reactions from the Reaxys database to estimate the number of reaction steps required to produce a molecule[59]. For both SAscore and SCscore, the higher the values are, the more difficult it is to synthesize the linker, hence less desirable. Specifically, the SAscore[60] is defined as the difference of fragment score and complexity penalty:

$$\text{SAscore} = \text{fragmentScore} - \text{complexityPenalty},$$

where fragment score is calculated by averaging over the contributions of non-zero elements of the Morgan fingerprint for all molecular fragments. The complexity score is calculated based on five components:

$$\text{complexityPenalty} = -\text{sizePenalty} - \text{stereoPenalty}$$
$$- \text{spiroPenalty} - \text{bridgePenalty} - \text{macrocyclePenalty}.$$

The size penalty increases with the number of atoms:

$$\text{sizePenalty} = \text{nAtoms}^{1.005} - \text{nAtoms},$$

The stereo penalty is calculated based on the number of chiral centers:

$$\text{stereoPenalty} = \log(\text{nChiralCenters} + 1),$$

The spiro penalty is calculated based on the number of spiro centers, or atoms that connect two rings together:

$$\text{spiroPenalty} = \log(\text{nSpiro} + 1),$$

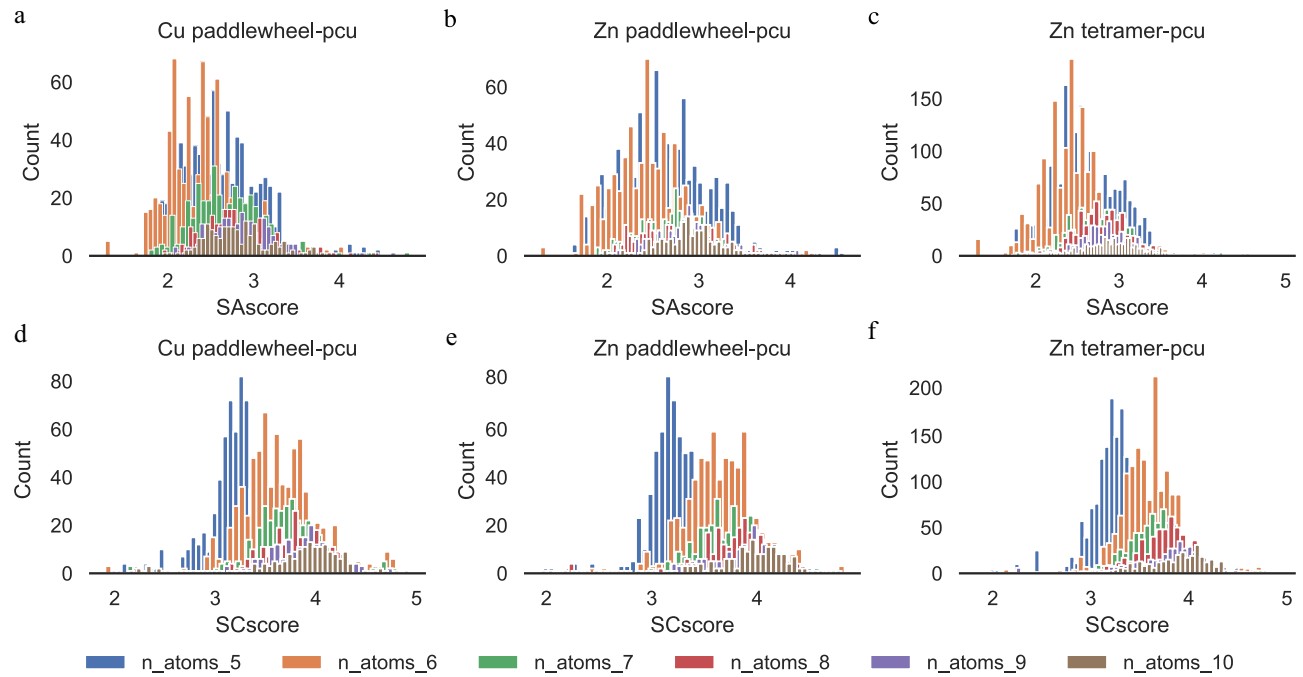

**Fig. 9 Synthesizability of DiffLinker-generated MOF linkers with dummy atoms. a–c** Distributions of synthetic accessibility score (SAscore); and **d–f** synthetic complexity score (SCscore) of DiffLinker-generated MOF linkers with dummy atoms. We indicate both node-topology pairs, and the number of sampled atoms, from 5 to 10 inclusive.

The bridge penalty term is calculated based on the number of bridgehead atoms, or atoms that belong to two or more rings:

$$\mathrm{bridgePenalty} = \log(\mathrm{nBridgeheads} + 1),$$

Lastly, if macrocycles are present, the macroPenalty is:

$$\mathrm{macrocyclePenalty} = \mathrm{math.}\log 10(2),$$

otherwise, the macroPenalty is zero. The SCscore[60] is calculated by using a neural network trained on 12 million reactions from the Reaxys database. The synthetic complexity of each molecule is rated from 1 to 5, with 1 being the easiest and 5 the most difficult to synthesize. To evaluate the capability of GHP-MOFassemble to generate a valid, novel, and diverse set of linkers, we leverage the MOSES framework[61] to compute three additional metrics: the fraction of valid linker SMILES strings (validity), the fraction of unique linker SMILES strings (uniqueness), and the dissimilarity of linkers (internal diversity). Validity measures whether atoms in generated molecules have the correct valency, whereas uniqueness quantifies the percentage of any molecule that is different from the rest of molecules.

We show in Fig. 9 the synthetic accessibility score (SAscore) and synthetic complexity score (SCscore) values for the remaining linkers. We observe that as the number of sampled atoms increases, both distributions generally shift to the right, with the exception of the SAscore distribution with six sampled atoms, which is to the left of that with five atoms. This general trend indicates that linkers become harder to synthesize as the molecules become bigger. This result is expected because as more atoms are sampled, more complex substructures may be present. We note that no linker has zero or very large SAscore or SCscore, values that would indicate unsynthesizability.

We show in Table 4 the validity, uniqueness, and internal diversity metrics, grouped by the number of sampled atoms and node-topology pairs of the corresponding MOFs. The validity column confirms that all generated linkers are valid. The last three columns, when reviewed from top to bottom, reveal a considerable increase in linker uniqueness and a more modest

increase in internal diversity as the number of sampled atoms increases. High uniqueness values indicate that our model is generating non-duplicate molecules, whereas high internal diversity values indicate that the generated molecules are chemically diverse. Internal diversity (which indicates how dissimilar a specific linker is to the rest of the population) is computed using two internal diversity scores: $\mathrm{IntDiv}_1$ and $\mathrm{IntDiv}_2$[61], also shown in Table 4. The increase in these metrics as more atoms are sampled is expected because the degrees of freedom of atomic species and their spatial coordinates increase with molecular size.

The internal diversity scores were calculated based on the Tanimoto distances[62] among all pairs of molecules, which are obtained by calculating the normalized Jaccard score of Morgan fingerprint bit vector between all pairs of AI-generated linkers. This yields similarities between a specific linker and the rest of the linkers in the linker pool, which are averaged out. Therefore, we end up with a metric showing each linker's similarity compared with the population of generated linkers. We used the MOSES[61] framework to compute the internal diversity scores $\mathrm{IntDiv}_1$ and $\mathrm{IntDiv}_2$[61] by using the relations:

$$\mathrm{IntDiv}_1(G) = 1 - \frac{1}{|G|^2} \sum_{m_1, m_2 \in G} T_d(m_1, m_2), \qquad (7)$$

$$\mathrm{IntDiv}_2(G) = 1 - \sqrt{\frac{1}{|G|^2} \sum_{m_1, m_2 \in G} T_d(m_1, m_2)^2}, \qquad (8)$$

where $T_d$ is the Tanimoto distance, which relates to the Tanimoto similarity ($T_s$) by:

$$T_d(m_1, m_2) = 1 - T_s(m_1, m_2), \qquad (9)$$

where $G$ is the generated set of molecules, $|G|$ is the size of that set, and $(m_1, m_2)$ is a pair of molecules in the set.

To measure the similarity between the AI-generated linkers and hMOF linkers in high-performing MOF structures, we show in Fig. 10 the distribution of maximum Tanimoto similarity

**Table 4 Statistical properties of AI-generated linkers.**

| node-topology | $\mathcal{N}$ | n_linker | $\mathcal{C}$ | $\mathcal{H}$ | valid | unique | IntDiv$_1$ | IntDiv$_2$ |
|---|---|---|---|---|---|---|---|---|
| Cu paddlewheel-pcu | 5 | 1240 | 774 | 466 | 1 | 0.496 | 0.692 | 0.669 |
| Zn paddlewheel-pcu | 5 | 1184 | 775 | 409 | 1 | 0.505 | 0.709 | 0.686 |
| Zn tetramer-pcu | 5 | 1666 | 1666 | 0 | 1 | 0.514 | 0.681 | 0.659 |
| Cu paddlewheel-pcu | 6 | 1117 | 761 | 356 | 1 | 0.545 | 0.718 | 0.695 |
| Zn paddlewheel-pcu | 6 | 992 | 681 | 311 | 1 | 0.554 | 0.718 | 0.697 |
| Zn tetramer-pcu | 6 | 1532 | 1532 | 0 | 1 | 0.512 | 0.693 | 0.672 |
| Cu paddlewheel-pcu | 7 | 499 | 383 | 116 | 1 | 0.919 | 0.751 | 0.734 |
| Zn paddlewheel-pcu | 7 | 453 | 329 | 124 | 1 | 0.933 | 0.758 | 0.742 |
| Zn tetramer-pcu | 7 | 709 | 709 | 0 | 1 | 0.866 | 0.744 | 0.729 |
| Cu paddlewheel-pcu | 8 | 330 | 245 | 85 | 1 | 0.947 | 0.759 | 0.742 |
| Zn paddlewheel-pcu | 8 | 338 | 247 | 91 | 1 | 0.943 | 0.763 | 0.745 |
| Zn tetramer-pcu | 8 | 550 | 550 | 0 | 1 | 0.905 | 0.746 | 0.732 |
| Cu paddlewheel-pcu | 9 | 281 | 191 | 90 | 1 | 0.979 | 0.756 | 0.739 |
| Zn paddlewheel-pcu | 9 | 257 | 175 | 82 | 1 | 0.989 | 0.763 | 0.746 |
| Zn tetramer-pcu | 9 | 389 | 389 | 0 | 1 | 0.959 | 0.742 | 0.727 |
| Cu paddlewheel-pcu | 10 | 235 | 165 | 70 | 1 | 0.988 | 0.769 | 0.750 |
| Zn paddlewheel-pcu | 10 | 217 | 148 | 69 | 1 | 1.000 | 0.768 | 0.749 |
| Zn tetramer-pcu | 10 | 316 | 316 | 0 | 1 | 0.994 | 0.750 | 0.737 |

Statistics of AI-generated linkers that correspond to different MOF types (first column), number of sampled atoms ($\mathcal{N}$ in the second column), number of carboxyl linkers ($\mathcal{C}$, fourth column), and number of heterocyclic linkers ($\mathcal{H}$, fifth column). In total, there are 12,305 linkers (see Table 2's Total column, Element filter row) with identified dummy atoms.

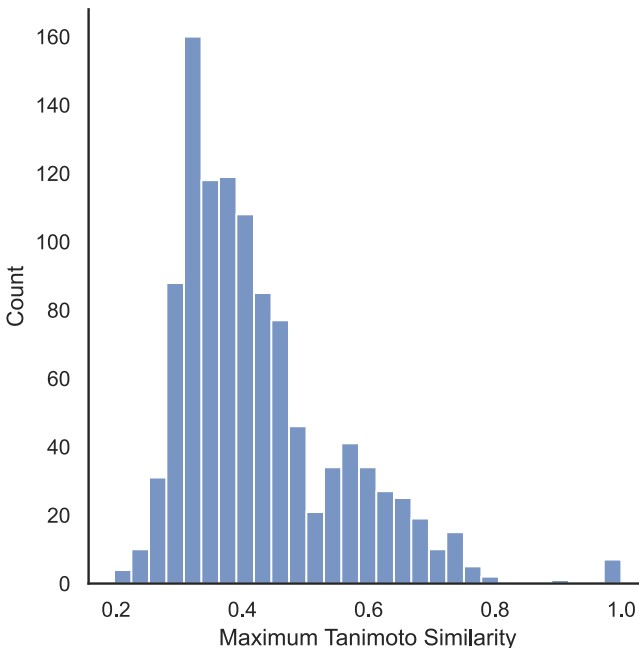

**Fig. 10 Similarity between AI-generated and hMOF linkers in high-performing MOF structures.** Distribution of maximum Tanimono similarity, which measures the uniqueness of AI-generated linkers as compared to hMOF linkers for high-performing MOFs. The peak around 0.3 to 0.4 indicates that most generated linkers are just 30–40% similar to those in hMOF, i.e., our AI framework generates novel linkers not present in hMOF structures. On the other hand, the trailing heavy right tail above 0.4 Tanimoto similarity indicates that we are also able to generate linkers that are structurally similar to those present in hMOF, showing that GHP-MOFassemble enables generation of a diverse set of novel linkers.

between the two linker sets. For each unique linker in the 364 AI-generated high-performing MOF structures we calculated its Tanimono similarity with all of the unique linkers in the hMOF structures. The maximum value of the Tanimoto similarities gives a quantitative measure of how different the generated linkers are from the hMOF linkers. Results in Fig. 10 indicate that most

linkers in the predicted high-performing MOF structures are vastly different from those in hMOF, i.e., our GHP-MOFassemble framework generates novel MOF structures with chemically unique linkers.

*MOF Assembly*. Once we have generated linkers, we can then assemble them with metal nodes. To construct MOFs, we need to guide the assembly process. We do this by using dummy atoms, which indicate the points at which the building blocks are to be connected. In practice, this is done as follows. Our parsing of MOFids generates, for MOFs with Zn tetramer nodes, linkers that carry two carboxyl groups, which by definition are part of metal nodes instead of linkers. Such wrong assignment of carboxyl groups is due to how the MOFid algorithm parses MOF structures. To reflect the correct molecular structure of linkers, the two carbon atoms in the carboxyl groups are identified as dummy atoms, and the redundant four oxygen atoms and two hydrogen atoms are removed. An illustration of dummy atom identification for linkers containing carboxyl groups is shown in the left panel of Fig. 11.

For MOFs with Cu paddlewheel and Zn paddlewheel node, however, another type of linker containing heterocyclic rings exists. For this type of linker, the two atoms that are nitrogen-metal bond distance away from the terminal nitrogen atoms on the heterocyclic rings are identified as dummy atoms. After the dummy atoms are correctly identified, three randomly selected linkers (duplicates allowed) are assembled with one of the three pre-selected nodes into MOFs in the pcu topology. An illustration of dummy atom identification of linkers containing heterocyclic groups is shown in the right panel of Fig. 11.

More than half of the hMOF structures are catenated MOFs, i.e., MOFs with interpenetrated lattices. By varying the level of interpenetration, it is possible to generate MOFs with different pore sizes, with a higher catenation level generally corresponding to smaller pores. We denote the four catenation levels in hMOF, with increasing number of interpenetrated lattices, as cat0, cat1, cat2 and cat3. To generate MOFs with high structural diversity, we applied site translation method as implemented in Pymatgen[63] to generate MOFs with different catenation levels. Figure 12 presents the building block combinations of the final six MOF candidates that passed all screening processes. For linkers,

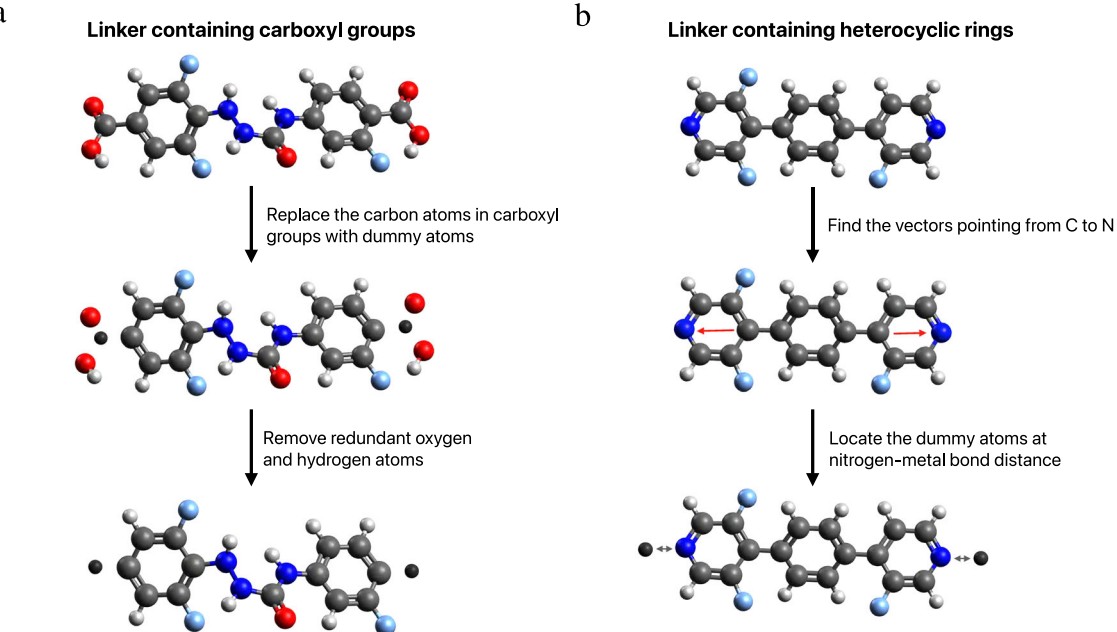

**Fig. 11 Assembly of AI-generated linkers with metal nodes. a** Identification of dummy atoms for linkers containing carboxyl groups. The dummy atoms are found by substituting the carbon atoms in the carboxyl groups. The remaining oxygen and hydrogen atoms in the carboxyl groups are removed. **b** Identification of dummy atoms for linkers containing heterocyclic rings. The dummy atoms are found at nitrogen-metal bond distance from the terminal nitrogen atoms along the vectors pointing from the opposing carbon atoms to nitrogen atoms.

| MOF ranking | 1 | 2 | 3 | 4 | 5 | 6 |
|---|---|---|---|---|---|---|
| Node | Cu PW | Zn TM | Cu PW | Cu PW | Zn PW | Cu PW |
| Linker 1 | | | | | | |
| Linker 2 | | | | | | |
| Linker 3 | | | | | | |
| Catenation | cat2 | cat3 | cat3 | cat2 | cat2 | cat2 |

**Fig. 12 Optimal linkers and catenation levels for MOF assembly.** Building blocks and catenation levels of the top six AI-generated MOF candidates.

the corresponding molecules (without dummy atoms) are shown for ease of visualization. For catenated MOF structures, we keep the spacing between interpenetrated lattices the same, so as to ensure equal pore sizes.

**Screening and predicting properties of AI-generated assembled MOFs.** Here we describe the final **Screen and Predict** component of GHP-MOFassemble. After the MOF structures are assembled, a comprehensive screening workflow is

developed to sift through the candidates with increasing levels of computational cost and confidence level:

- Inter-atomic Distance Check (screen)
- Pre-simulation Check (screen)
- Predicting $CO_2$ Capacity of MOF via Pre-trained Regression Model (screen and predict)
- Structure Validation of the Assembled MOFs (screen)
- Grand Canonical Monte Carlo Simulation (screen and predict)

**Table 5 Statistical properties of AI ensemble to characterize MOF's performance.**

| Model | $R^2$ | MAE | RMSE |
|---|---|---|---|
| Model1 | 0.932 | 0.098 | 0.171 |
| Model2 | 0.937 | 0.100 | 0.170 |
| Model3 | 0.936 | 0.099 | 0.170 |

Statistics of AI ensemble, including $R^2$ score, mean absolute error (MAE), and root mean squared error (RMSE).

**Table 6 Catenation level of high-performing MOFs.**

| node-topology | cat0 | cat1 | cat2 | cat3 |
|---|---|---|---|---|
| Cu paddlewheel-pcu | 0 | 1 | 38 | 26 |
| Zn paddlewheel-pcu | 0 | 0 | 66 | 23 |
| Zn tetramer-pcu | 0 | 0 | 53 | 157 |

Numbers of predicted high-performing MOFs found in the hMOF dataset.

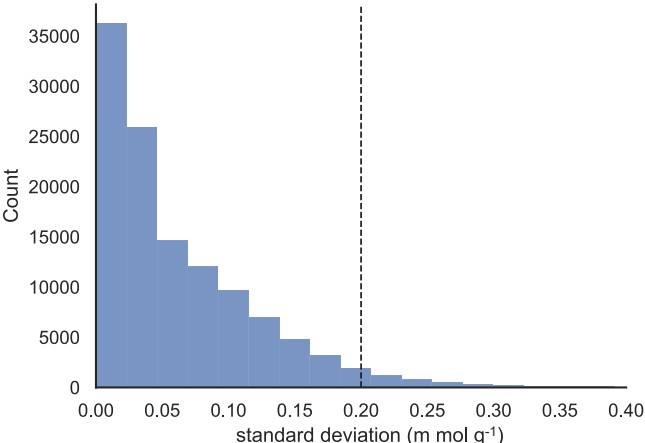

**Fig. 13 Standard deviation of AI model ensemble to estimate MOFs' $CO_2$ capacity.** Distribution of the standard deviation of AI ensemble predictions. Predictions with standard deviation of less than 0.2 m mol g$^{-1}$ consist of around 96% of the test set. This shows that our three independently trained AI models are in great agreement. The remaining 4% with larger than 0.2 m mol g$^{-1}$ standard deviation implies that these are the data points which may be difficult to predict due to errors in target property or extra information other than atomic species and periodic neighbor being necessary for accurate predictions.

*Inter-atomic Distance Check.* A pairwise distance matrix $M_{n \times n}$ is extracted from a MOF's CIF file. The matrix element $M_{i,j}$ denoting the Euclidean distance between atom $A_i$ and atom $A_j$ is:

$$M_{i,j} = \sqrt{(x_i - x_j)^2 + (y_i - y_j)^2 + (z_i - z_j)^2}, \quad (10)$$

where $1 \le i \le n$ and $1 \le j \le n$, and $(x_i, y_i, z_i)$ and $(x_j, y_j, z_j)$ are the Cartesian coordinates of atoms $A_i$ and $A_j$. For an arbitrary pair of atoms $A_i$ and $A_j$, their chemical symbols can be denoted as $E_i$ and $E_j$, respectively. The bond length between atoms $A_i$ and $A_j$ is $\delta(E_i, E_j)$. The minimum allowed inter-atomic distance between these two atoms can be estimated by calculating the infimum of all experimental bond lengths between these two element types, i.e.: $\inf(\delta(E_i, E_j))$, which can be queried from the OChemDb[38] database. All values of $M_{i,j}$ $(i \ne j)$ are compared against $\inf(\delta(E_i, E_j))$, and any occurrence of $M_{i,j} < \inf(\delta(E_i, E_j))$ $(i \ne j)$ will result in the MOF candidate being discarded. Entries with $i = j$ (i.e., the diagonal entries of the distance matrix $M_{i,i}$) are not examined as they should always be zero, because the distance between an atom and itself is zero. We used Python 3.10's built-in multiprocessing library to help launch the inter-atomic distance check on different MOF structures in parallel.

*Pre-simulation Check.* For each atom in the MOF structure, the pre-simulation check workflow can be described as the following:

1. The atom's element should be recognized as one of the supported elements in the UFF4MOF parameter set,
2. The atom's neighboring atoms that are within the allowed bonding distance are examined for chemistry validity. We conducted this process with the help of the cif2lammps library. Python's multiprocessing library is also used to speed up this process over many structures in parallel.

*Predicting $CO_2$ capacity of MOF via pre-trained regression model.* The GHP-MOFassemble framework uses an ensemble of regression models to estimate the $CO_2$ capacity of newly generated pcu MOF structures. We use for this purpose a modified version of the CGCNN model, developed in our previous work[37], which adopts an adjacency list to format node and edge embeddings, rather than the adjacency matrix format of the original CGCNN, a change that enhances model training speed, training stability, and prediction accuracy. To fine-tune this AI model, we used MOF structures in the hMOF dataset, along with their $CO_2$ capacities at 0.1 bar as input. We split the hMOF dataset into 80% training set, 10% validation set, and 10% test test. We independently trained three modified versions of the CGCNN model for 5000 epochs with a batch size of 160, using Adam as the optimizer, at a learning rate of $10^{-4}$, and a weight decay rate of $2 \times 10^{-5}$.

Table 5 shows the $R^2$ score, mean absolute error (MAE), and root mean squared error (RMSE) of the three AI models ensemble on the 10% test set. Figure 13 shows the distribution of the standard deviations of ensemble model predictions. We treat the standard deviation of predictions from the three models as a measure of model uncertainty, which we view as arising from the model's difficulty in learning certain data points. This uncertainty is also known as epistemic uncertainty. By employing an ensemble of models and averaging their prediction results, we can increase prediction accuracy, which is thoroughly and extensively explored in the machine learning community[64,65]. We chose the threshold for the standard deviation filter to be 0.2 m mol g$^{-1}$ because it is sufficiently small (with 96% of data points below the threshold) that for low $CO_2$ capacity predictions, the predicted values across the ensemble are similar. On the other hand, for high $CO_2$ capacity predictions, it also ensures that the outliers (i.e., predicted values across the ensemble are vastly different) are filtered out, therefore minimizing the overall error. In Table 6 we also notice that most of the high-performing MOFs are cat2 and cat3. This is consistent with our preliminary analysis on catenation levels and distributions of $CO_2$ adsorption capacity in Fig. 5.

We show in the left panel of Fig. 14 the scatter plot of the ensemble model predictions for the 10% test set (from hMOF), which has a MAE of 0.093 m mol g$^{-1}$. Using $CO_2$ capacity of 2 m mol g$^{-1}$ as a threshold, we repurposed our predictive model as a classifier to categorize MOFs into low and high performers, with predicted $CO_2$ capacities below and above the threshold, respectively. Using this scheme, the confusion matrix for identifying low and high performers is shown in the right panel of Fig. 14. We conclude from the confusion matrix that the pre-trained model classifies both low and high performers with high

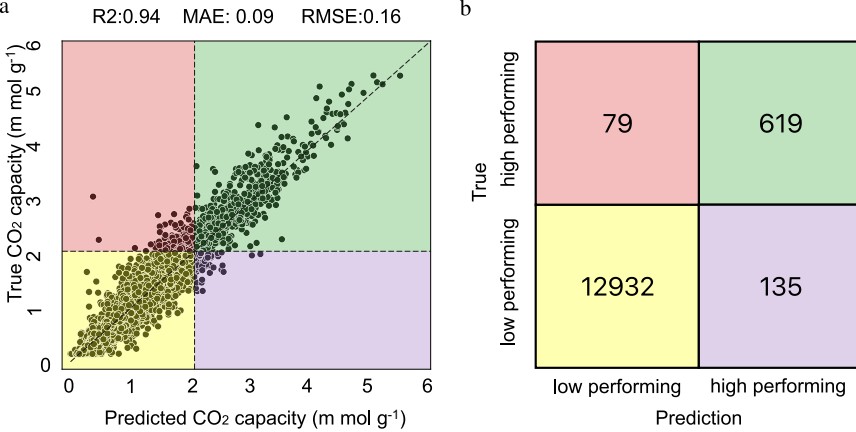

**Fig. 14 Performance of AI ensemble to measure CO$_2$ capacity. a** Predictive performance of the ensemble model on the 10% test set of hMOF. **b** Confusion matrix of the ensemble model in classifying low and high performers in the test set. MOFs with real/predicted CO$_2$ capacity lower than 2 m mol g$^{-1}$ are identified as low performers, and MOFs with real/predicted CO$_2$ capacity higher than 2 m mol g$^{-1}$ are identified as high performers.

accuracy, with 98.4% (13,551 out of 13,765) of test samples (of hMOF) correctly classified. As the test set is heavily imbalanced, with many more low performers than high performers, we also calculated the balanced accuracy of classification[66] (the sum of true positive rate and true negative rate divided by two for binary classification or average of recalls for multi-class classification), obtaining a value of 90.7%. Since the majority of the 214 misclassified samples lie close to the decision boundaries, as shown in the red and purple regions of the left panel of Fig. 14, we conclude that the ensemble model is capable of differentiating low and high performers.

We show in Fig. 15 the empirical cumulative distribution functions of the CO$_2$ capacities of hMOF (dashed lines) structures and AI-generated MOF structures (solid lines) at different catenation levels. We observe that there are more high-performing cat2 and cat3 MOFs among the generated structures compared to the hMOF dataset, as indicated by lower cumulative density values at 2 m mol g$^{-1}$ (black dashed line). On the other hand, there are fewer high-performing cat0 and cat1 MOFs, as shown by higher cumulative density values at 2 m mol g$^{-1}$. This result demonstrates that within the AI-generated MOF design space, there are more high-performing candidates with higher catenation levels compared to lower catenation levels.

In summary, we use an ensemble of 3 AI models to identify pcu AI-generated MOFs produced by the **Generate** step (in addition to inter-atomic distance and pre-simulation checks) that have predicted CO$_2$ capacities higher than 2 m mol g$^{-1}$. We use the ensemble mean *plus* standard deviation as the final prediction. These MOFs are then selected for further investigation with MD simulations.

*Structure Validation of the Assembled MOFs.* Pre-screened MOFs are checked for stability using MD simulations with LAMMPS[42], version stable release 2 August 2023 compiled with gcc 11.2.0 and OpenMPI 4.1.2. For each MOF, a 2x2x2 supercell of triclinic periodic boundary condition is created. A triclinic isothermal-isobaric ensemble (i.e., NPT) at 300 K and 1 atm is applied to the supercell structure. All lattice parameters: $a$, $b$, $c$, $\alpha$, $\beta$, $\gamma$ are allowed to equilibrate freely. A relatively short simulation with 400,000 steps is conducted for each MOF to allow for structural equilibrium with a step size of 0.5 femtoseconds. The potential energy $E_{pot}$ of the MOF structure can be expressed as:

$$E_{pot} = E_{bond} + E_{angle} + E_{dihedral} + E_{improper} + E_{Vdw} + E_{Coul}, \quad (11)$$

where $E_{bond}$ is the bond stretch energy, $E_{angle}$ is the angle between bonds energy, $E_{dihedral}$ is the dihedral angle energy, $E_{improper}$ is the improper dihedral angle energy, $E_{vdw}$ is the nonbonded interaction energy, and $E_{coul}$ is the Coulombic interaction energy.

Coupry et al. (2016)[40] stated that 76.5% of the CoRE structures[14] lattice parameters are within 5% change when they are simulated with the UFF4MOF parameters. A similar criterion is used here such that if any of MOF's lattice parameters of triclinic cell changes are more than 5%, the structure is discarded in the screening process. This screening process provides a new level of confidence in terms of structural stability of the AI-generated MOFs. Figure 16 demonstrates the relative error of all lattice parameters of the top six MOF candidates during the MD simulation.

*Grand Canonical Monte Carlo (GCMC) Simulations.* The void fraction of each candidate MOF is calculated using the RASPA[43]'s helium void fraction function. It helps us understand the porosity contribution toward MOF CO$_2$ adsorption capacity. Next, PACMOF[67] is used to assign partial charges to MOFs. Using the pre-trained partial atomic charge model on the density-derived electrostatic and chemical (DDEC) charge[68] data, the partial charges are assigned to atoms in MOF structures. After determining the partial charges, the MOF structures are assigned with UFF4MOF force field parameters. The GCMC simulations are conducted at 0.1 bar and 300 K with a step size of 0.5 fs/step. The MOF structures are under the rigid assumption, therefore only Van der Waals interactions and Coulombic interactions (i.e., non-bonded) are considered:

$$E_{pot} = E_{Vdw} + E_{Coul}, \quad (12)$$

which can be expressed as[69]

$$E_{pot} = \sum_{i,j} 4\epsilon_{ij} \left[ \left( \frac{\sigma_{ij}}{r_{ij}} \right)^{12} + \left( \frac{\sigma_{ij}}{r_{ij}} \right)^{6} \right] + \frac{q_i q_j}{4\pi\epsilon_0 r_{ij}}, \quad (13)$$

where $\epsilon_{ij}$, $\sigma_{ij}$, and $r_{ij}$ are the well depth, zero position, and interatomic distance of a Lennard-Jones 12-6 interaction between atom $i$ and atom $j$; $q_i$ and $q_j$ are the electrostatic charges of atom $i$ and atom $j$; $\epsilon_0$ is the vacuum permittivity.

**Computational resources and performance.** We have deployed and extensively tested GHP-MOFassemble on computers at the ALCF and at the National Center for Supercomputing Applications (NCSA), with the intent of providing scalable and

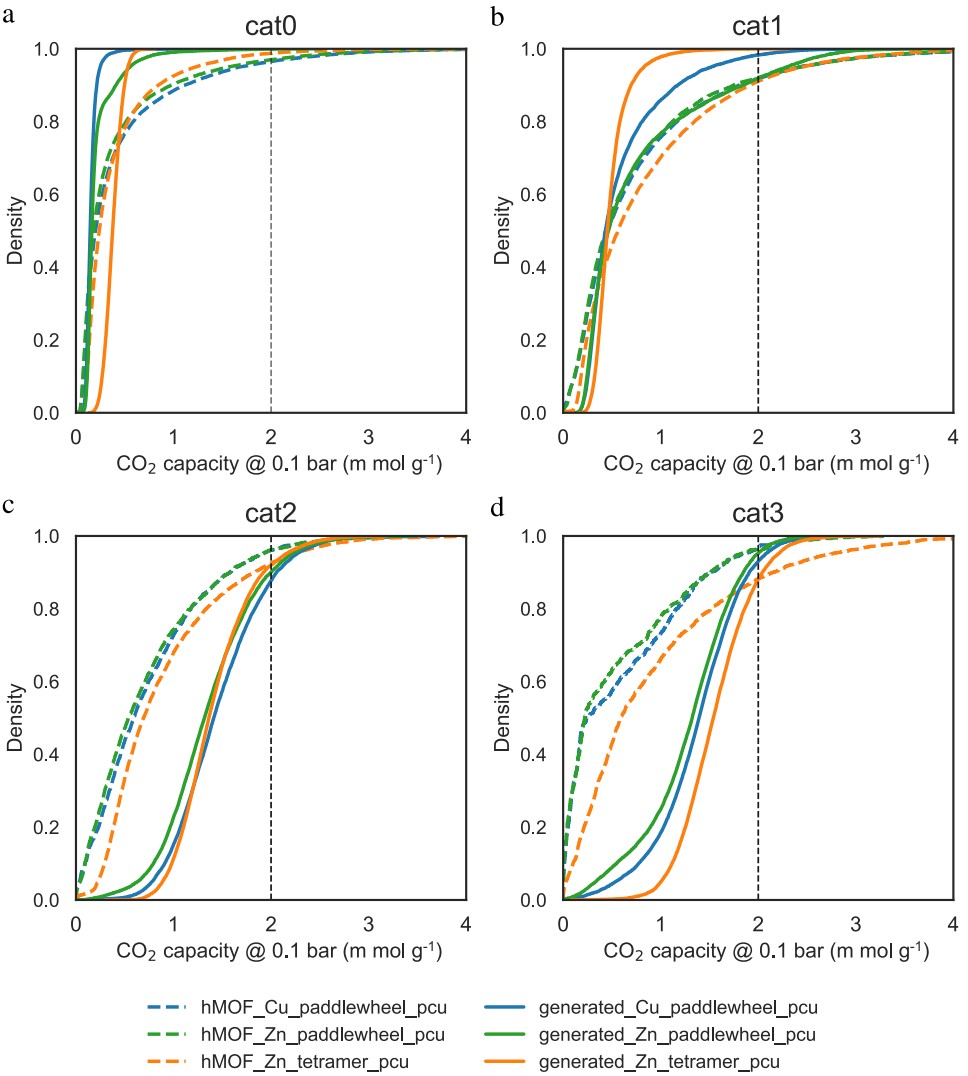

**Fig. 15 MOFs' CO$_2$ capacities in terms of catenation levels.** Comparison of empirical cumulative distribution functions of the predicted CO$_2$ capacities of generated and hMOF structures for **a** cat0, **b** cat 1, **c** cat2, and **d** cat 3.

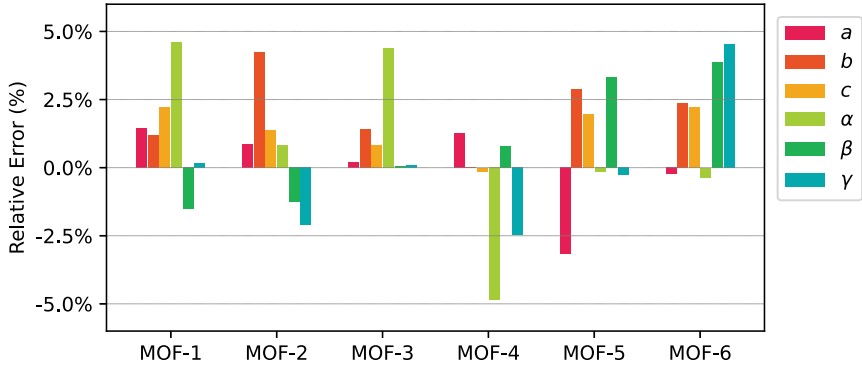

**Fig. 16 Lattice parameter relative errors of the top 6 AI-generated MOFs.** The top six AI-generated MOF candidates have change less than 5% in all lattice parameter during MD simulation. The first three lattice parameters are cell axes length and the latter three lattice parameters are cell plane angles. Their CO$_2$ capacities are higher than 96.9% of MOF structures in the hMOF database.

computationally efficient AI tools to accelerate the modeling and discovery of novel MOF structures. The tools introduced in this work may be readily fine-tuned and adapted to other available datasets beyond hMOF to enable accelerated design and discovery of novel MOF structures for carbon capture at industrial scale.

The training and inference of the generative model and the training of the regression model was conducted on 8-way NVI-DIA A100 GPUs with FP32 mode. The inference of the regression model is conducted on a single NVIDIA A40 GPU with FP32 mode. CPU-based screening processes, including distance check,

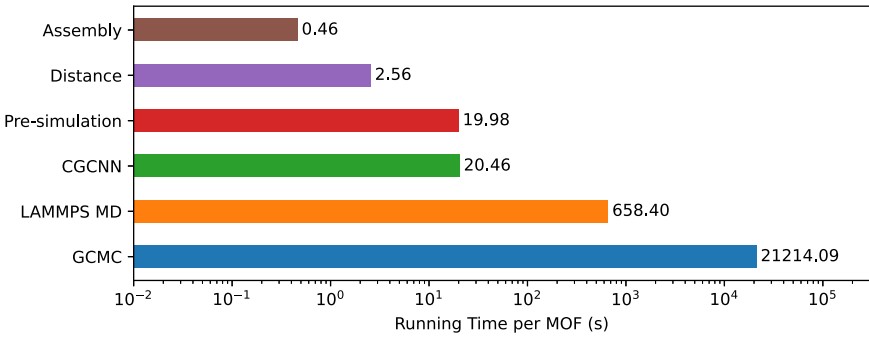

**Fig. 17 Process time cost breakdown of GHP-MOFassemble.** Benchmark analysis of the computational cost for each process of the GHP-MOFassemble framework. Notice that the AI component is hyperefficient.

pre-simulation check, MD simulations, and GCMC simulations, are run on two-way AMD Epyc 7763 CPUs. We now present a breakdown of the average computational cost of each step of the GHP-MOFassemble workflow per MOF:

- MOF assembly: 0.46 seconds per core. We scaled up this analysis by multiprocessing on 28 cores.
- Distance check to ensure structural validity: 2.56 seconds per core. We scaled up this analysis by multiprocessing on 128 cores.
- Pre-simulation check to ensure chemical consistency: 19.98 seconds per core. We scaled up this analysis by multiprocessing on 128 cores.
- AI ensemble inference of $CO_2$ adsorption capacity: 20.46 seconds. We used one NVIDIA A40 GPU.
- MD simulations to validate stability and chemical consistency: 658.40 seconds, or about ∼ 10 minutes. We scaled up this analysis using between six to 14 MPI processes based on the number of atoms in the MOF.
- Detailed GCMC simulations: 21214.04 seconds, or about ∼6 hours. This averaged number is based on simulations on one CPU core.

A schematic representation of this benchmark analysis is presented in Fig. 17.

## Data availability

The datasets generated during and/or analysed during the current study are available in the GitHub repository, https://github.com/hyunp2/ghp_mof/tree/main/utils. We also used the open source hMOF dataset[70], and the GEOM dataset[56].

## Code availability

The scientific software and data used in this article are readily available in `GitHub` at https://github.com/hyunp2/ghp_mof/tree/main.

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

## Acknowledgements

This work was supported by Laboratory Directed Research and Development (LDRD) funding from Argonne National Laboratory, provided by the Director, Office of Science, of the U.S. Department of Energy under Contract No. DE-AC02-06CH11357, and by the Braid project of the U.S. Department of Energy, Office of Science, Advanced Scientific Computing Research, under contract number DE-AC02-06CH11357. The work used resources of the Argonne Leadership Computing Facility, a DOE Office of Science User Facility supported under Contract DE-AC02-06CH11357. EAH and IF acknowledge support from National Science Foundation (NSF) award OAC-2209892. SC and XY acknowledge partial support from NSF Future of Manufacturing Research Grant 2037026. This research also used the Delta advanced computing and data resources, which is supported by the National Science Foundation (award OAC 2005572) and the State of Illinois. Delta is a joint effort of the University of Illinois at Urbana-Champaign and its National Center for Supercomputing Applications.

## Author contributions

E.A.H. envisioned and led this work, and guided the connection between generative AI with MD and GCMC simulations to create novel, chemically consistent MOFs at scale in high performance computing platforms. H.P. adapted techniques from drug design into MOF discovery with generative AI, producing a pool of candidates of MOF components, and also trained and inferred assembled MOFs' CO2 adsorption capabilities using a graph neural network model. X.Y. developed codes to decompose MOF into linkers and nodes, as well as to assemble predicted linkers and nodes to make hypothetical MOFs, and also conducted large-scale MD simulations, with LAMMPS and GCMC to identify stable and synthesizable AI-predicted MOFs. R.Z. analyzed hMOF database to understand MOF composition statistics, tested combined generative AI and graph models to predict novel AI-generated MOFs and quantify their chemical properties. S.C. provided expertise on metrics to be used to study the chemical properties and stability of MOFs with AI tools developed in this manuscript. D.C. provided expertise on expected features

of MOFs to be used for carbon capture at industrial scale. I.F. guided the development of our ready-to-use AI framework in modern computing environments, and provided guidance on how to use and interpret metrics for MOF synthesizability. E.T. provided expertise on MD, chemistry and synthesis pathways of MOFs as well as how to analyze MD results. All authors reviewed and contributed to the manuscript.

## Competing interests
The authors declare no competing interests.
