## [Peer Review File · Communications Chemistry]

Reviewers' comments:

Reviewer #1 (Remarks to the Author):

In this manuscript, Huerta et al. propose a new methodology to generate unexplored pcu-topology MOFs and find the materials with high CO₂ capacity and synthesizable linkers. To generate new MOFs, they used a diffusion-based generative model called DiffLinker that was originally designed to obtain chemically relevant candidate drug molecules. These linkers are then assembled into MOF structures using different levels of catenation. These newly designed MOFs are then evaluated using a neural network model previously developed by the same group. After generating 120,000 pcu MOFs using their method, they found that only 102 structures with a CO₂ capacity higher than 2mmol/g at 0.1 bar could complete a MD simulation at 1 bar. Among them only 18 can be considered stable.

I find the approach very interesting since it uses state-of-the-art ML models to generate and evaluate MOFs for carbon capture. The article is very well written and easy to follow, and I learned a lot on diffusion models. This type of study is quite new since it relies on generative modeling borrowed to the latest advance in data science, which stands out from the more standard high-throughput screening methodologies. However, I find that the manuscript lacks practical perspectives and chemical insight on the results of the generative modeling. The authors can for instance analyze the 18 MOFs found by their method and evaluate their adsorption performance using the more reliable GCMC calculations instead of the result of the ML predictions as a validation. Moreover, the chemical nature of the linkers is very quickly tackled while more analysis could be made. For these reasons, I suggest publication after minor revisions.

Major comments

The final section on validation could be further enriched by selecting some structures from the 18 MOFs (the most promising ones) and by carrying out a more standard performance screening and by clearly specifying their structures. The authors can carry out a similar approach as in the work on SmVAE cited in ref 28. This study can be done for example on the 5 structures shown in Figure 10.

The manuscript lacks a bit of chemical insight. For example, the authors could discuss the groups present in the linkers of the most promising materials, both in the input data (hMOF) and output dataset generated by the DiffLinker. Are they polar groups? or aromatics? What is their length?

Page 6: "... resulted in novel and chemically diverse MOF linkers". Please specify how the novelty and chemical diversity is measured.

Page 7: the variables are defined but the functions q , p and N are not defined at all. Please refer to ref 14 for a clearer definition of these functions.

Page 8: "Since the pre-trained DiffLinker model we used in this work was trained on the GEOM dataset, ..." The causality is not clear to people who does not know the dataset. Please clarify the underlying

reason.

The SAscore and SCscore are not clearly defined in the manuscript even references are given. I would suggest adding a definition in the appendix like the other 3 metrics used to evaluate the linkers. Moreover, the analysis of the results on SA and SC scores becomes even more obvious in the light of the definition since these scores considers the number of molecules in the scoring (if I understood correctly).

In Figure 2, 5162 linkers could potentially generate 137 billion of structures if the linkers are drawn with replacement (and if no catenation is considered). It is not clear how the 40,000 MOFs are selected among them (is it just random selection, is there a bias process to ensure diversity). Please comment on this step of the algorithm.

On catenation, I would suggest more insight on why higher catenation level favor adsorption. Is it because of smaller pore size that increase the interactions with CO₂? Please add some discussions on this aspect by maybe calculating the pore size, pore volume and pore surface using Zeo++. A comparison of catenation level to structural properties would be much clearer than the discussion around Figure 8.

Page 14: "three times to create an ensemble of models". Please comment on this assertion. I would suggest putting the performance of the ensemble model in Table D1. Moreover, I did not understand how the three models are different from each other and how they decrease bias. To be effective, each model would need to be trained in a slightly different manner else, the three models are biased in the same way and the final model is also biased. Please clarify this point.

Minor comments

Some typos mainly.

Page 8: "manuallyfilter"

Page 10: "by, for example by"

Page 16: "(sold lines)"

Page 17, first few lines: repetition of "indicate"

Page 18: "phenylene et al."

The reproducibility of the work is appreciated since a Github repository is provided with the code and the data associated.

Reviewer #2 (Remarks to the Author):

The paper by Park and co-workers uses machine learning to generate MOF structures in an interesting and innovative way. I recommend publication, but would like to have a few simple questions answered first.

Their MD methodology wasn't 100% clear to me outside of the basics. They look to determine material stability with MD, but how did they come up with 1% density change rule? Is this standard for MOF NPT calculations? For clarity, they should do a light expansion (e.g., perhaps a paragraph) on the MD portion. Why is this specific route and their metrics used? Are they standard for MOFs? This might be obvious to some readers, but not necessarily to all.

Several related notes that I hope they can discuss. They clearly set a target for accelerated discovery of materials with ML, but the amount of computational chemistry calculations they have to do after the ML is obscene. The ML generated 6000 structures to study via computational chemistry, more than 98% of the structures must have been absurd to fully disintegrate in MD. The MD found less than 1% of the suggested structures stable (18 out of 6000), and only 102 out of 6000 structures (less than 2%) that were even remotely reasonable.

How useful is AI that requires you afterwards to do MD simulations on 6000 structures, and after the MD you find that the VAST majority of those generated structures were useless and unphysical?

These comments likely sound more negative than my intention, the final structures are quite interesting, and the procedure is logical and innovative. I don't think they need to solve these problems for publication. I do think though that they should at least more thoroughly discuss them, and better comment about the utility and scope of such an approach.

Response to the reviewers

We thank the reviewers for their constructive comments and suggestions. A detailed point-by-point response may be found below. New content in our revised manuscript is marked in red to facilitate its review.

Reviewers' comments:

Reviewer #1 (Remarks to the Author):

In this manuscript, Huerta et al. propose a new methodology to generate unexplored pcu-topology MOFs and find the materials with high CO₂ capacity and synthesizable linkers. To generate new MOFs, they used a diffusion-based generative model called DiffLinker that was originally designed to obtain chemically relevant candidate drug molecules. These linkers are then assembled into MOF structures using different levels of catenation. These newly designed MOFs are then evaluated using a neural network model previously developed by the same group. After generating 120,000 pcu MOFs using their method, they found that only 102 structures with a CO₂ capacity higher than 2mmol/g at 0.1 bar could complete a MD simulation at 1 bar. Among them only 18 can be considered stable.

I find the approach very interesting since it uses state-of-the-art ML models to generate and evaluate MOFs for carbon capture. The article is very well written and easy to follow, and I learned a lot on diffusion models. This type of study is quite new since it relies on generative modeling borrowed to the latest advance in data science, which stands out from the more standard high-throughput screening methodologies. However, I find that the manuscript lacks practical perspectives and chemical insight on the results of the generative modeling. The authors can for instance analyze the 18 MOFs found by their method and evaluate their adsorption performance using the more reliable GCMC calculations instead of the result of the ML predictions as a validation. Moreover, the chemical nature of the linkers is very quickly tackled while more analysis could be made. For these reasons, I suggest publication after minor revisions.

Response: Thank you for sharing these excellent suggestions! In our revised manuscript we have introduced several major improvements to address your comments, namely:

- We have included an additional step in our screening workflow which consists of checking that the minimal distance among atoms in AI-assembled MOFs follows the thresholds provided in the experimental database OChemDb. We optimized this part of the workflow using multiprocessing so that each MOF is screened within 0.02 seconds on 128 CPU cores.
- We used the open source library, cif2lammmps, and the UFF4MOF force field to generate LAMMPS input files. This step ensures that all atomic structures and bonds appearing in each AI-generated MOF structure are chemically valid. We optimized this part of the

workflow using multiprocessing, thereby examining each MOF within 0.15 seconds on 128 CPU cores.

- Once we have identified chemically valid MOFs, we used the CGCNN model to filter out all MOF structures with capacities less than 2 mmol/g. This process takes 0.16 seconds per MOF.
- After selecting AI-generated MOFs with a CO₂ capacity larger than 2 mmol/g, we used LAMMPS to examine their stability and porous properties. This process takes 11 minutes per MOF.
- MOFs that according to LAMMPS simulations have changes less than 5% in any of their lattice parameters were selected for further tests with GCMC simulations to calculate their CO₂ adsorption capacity at 0.1 bar. In our revised manuscript we present 6 novel MOFs with CO₂ capacity larger than 2 mmol/g. This process takes 6 hours per MOF.

Major comments

The final section on validation could be further enriched by selecting some structures from the 18 MOFs (the most promising ones) and by carrying out a more standard performance screening and by clearly specifying their structures. The authors can carry out a similar approach as in the work on SmVAE cited in ref 28. This study can be done for example on the 5 structures shown in Figure 10.

Response: This is an excellent recommendation. After revamping our screening and validation workflow, we identified 364 MOFs which, according to our CGCNN model, had CO₂ capacity larger than 2 mmol/g. We then studied their stability with LAMMPS, and found 102 stable MOFs with a CO₂ capacity larger than 2 mmol/g. We ran GCMC simulations on these 102 MOFs and identified 6 top performing MOFs with CO₂ capacities larger than 2 mmol/g, one of them with a mean CO₂ capacity of 3.686 mmol/g. We present these new results in the revised manuscript. In Methods, we present the computational cost of each of the analyses; see Figure 16.

The manuscript lacks a bit of chemical insight. For example, the authors could discuss the groups present in the linkers of the most promising materials, both in the input data (hMOF) and output dataset generated by the DiffLinker. Are they polar groups? or aromatics? What is their length?

Response: Apologies for this oversight. We added more in-depth analysis of functional groups of linkers in the AI-generated high-performing MOFs, and compared them to those in high-performing hMOFs. These new results can be seen at a glance in Figure 8 of the Results section. In other words, we find functional groups in linkers from both hMOF dataset and AI-generated MOFs that act as hydrogen donors to CO₂ molecules, as discussed in Ref. [38].

Page 6: "... resulted in novel and chemically diverse MOF linkers". Please specify how the novelty and chemical diversity is measured.

Response: We have rewritten our manuscript to ensure that the definition and use of these metrics is easy to follow by the reader. Thus, we removed the Appendices and put all the relevant information in the Methods section. See the revised subsection *Generating new MOF structures* in Methods, and in particular the discussion under the heading *Screen and Evaluate*.

Page 7: the variables are defined but the functions q , p and N are not defined at all. Please refer to ref 14 for a clearer definition of these functions.

Response: We added the definitions for (q , p , N) after Equation 6 and added the relevant citation.

Page 8: "Since the pre-trained DiffLinker model we used in this work was trained on the GEOM dataset, ..." The causality is not clear to people who does not know the dataset. Please clarify the underlying reason.

Response: We agree. A pre-trained version of the DiffLinker model is a generative model specialized in generating organic molecular fragments. In our generative approach, the DiffLinker's role is to sample new connecting components for fragments of each hMOF linker/ligand. We believe that the GEOM database is suitable to learn the statistical pattern of organic molecular fragments in general, due to high diversity of molecular types and conformations (i.e., 37 million molecular conformations for over 450,000 molecules). In this way, we are ensuring the generated linkers/ligands are always different from any existing linker/ligands within the hMOF structures, yet valid. We added the following information in the Methods section under *Generating new MOF structures > Diffuse and Denoise*:

The SAscore and SCscore are not clearly defined in the manuscript even references are given. I would suggest adding a definition in the appendix like the other 3 metrics used to evaluate the linkers. Moreover, the analysis of the results on SA and SC scores becomes even more obvious in the light of the definition since these scores consider the number of molecules in the scoring (if I understood correctly).

Response: Agreed. We have expanded and clarified the definition of these metrics in the Methods section. See specifically *Generating new MOF structures > Screen and Evaluate*.

In Figure 2, 5162 linkers could potentially generate 137 billion of structures if the linkers are drawn with replacement (and if no catenation is considered). It is not clear how the 40,000 MOFs are selected among them (is it just random selection, is there a bias process to ensure diversity). Please comment on this step of the algorithm.

Response: We constructed a total of 120,000 MOFs (for all three node types) by randomly selecting three linkers and assembling them with one of the three node types. The random selection of linkers is to reduce bias during selection. The total number of the generated MOFs (120,000) is chosen to ensure that it is comparable with the number of MOFs in the hMOF database.

On catenation, I would suggest more insight on why higher catenation level favor adsorption. Is it because of smaller pore size that increase the interactions with CO₂? Please add some discussions on this aspect by maybe calculating the pore size, pore volume and pore surface using Zeo++. A comparison of catenation level to structural properties would be much clearer than the discussion around Figure 8.

Response: This is an excellent point. We have expanded our discussion on this point both in the Results and Methods sections. Specifically, we have included this information in the Results section:

This result confirms that catenation is an important factor when designing new MOF structures with high CO₂ capacity. This observation is consistent with other studies in the literature [38,39], which indicate that even though catenation reduces the pore size and surface area, catenated MOFs generally have higher CO₂/H₂ selectivities because MOF-CO₂ interactions are enhanced as a result of the strong confinement of CO₂ with a much lower adsorption surface. Thus, the results presented in Figure 2 for the hMOF dataset, and for our AI-generated MOFs in Table 6, indicate that CO₂ working capacities of catenated MOFs are higher than their non-catenated counterparts.

Page 14: “three times to create an ensemble of models”. Please comment on this assertion. I would suggest putting the performance of the ensemble model in Table D1. Moreover, I did not understand how the three models are different from each other and how they decrease bias. To be effective, each model would need to be trained in a slightly different manner else, the three models are biased in the same way and the final model is also biased. Please clarify this point.

Response: We have included a new Results subsection titled *Regression model for MOF CO₂ capacity prediction* in which we provide this information. In brief, we trained, validated, and tested three independent CGCNN models using MOF structures from the hMOF dataset as well as their CO₂ capacities at 0.1 bar as input data. Using the same training, validation and test sets, architectures, and all other relevant hyperparameters, we used random initialization of weights to create three independent AI models. We present the R² score, mean absolute error (MAE), and root mean squared error (RMSE) of the 3 regression model ensemble on the test set. Then to infer the CO₂ capacities of newly generated MOF structures, we take the average of the predictions made by the three independent models as the predicted CO₂ adsorption capacity. This approach—ensemble learning—is extensively used in the ML/AI community to yield better performance on ML problems, such as regression—the central application of our AI ensemble, see e.g., <https://neptune.ai/blog/ensemble-learning-guide>. We have added this information in our revised manuscript. In addition, we make sure we discard highly uncertain predictions for further processing (by computing ensemble standard deviation). Such model uncertainty, known as epistemic uncertainty, is effective in filtering out predictions which do not have consensus. If we were to use only one AI model, we rely solely on one source of prediction, which would make our predictions less reliable.

Minor comments

Some typos mainly.

Page 8: “manuallyfilter”

Page 10: “by, for example by”

Page 16: “(sold lines)”

Page 17, first few lines: repetition of “indicate”

Page 18: “phenylene et al.”

Response: Thank you, we corrected all these typos and have proofread the manuscript again.

The reproducibility of the work is appreciated since a Github repository is provided with the code and the data associated.

Response: Thank you. All the improvements described in our revised manuscript are now included in this repository.

Reviewer #2 (Remarks to the Author):

The paper by Park and co-workers uses machine learning to generate MOF structures in an interesting and innovative way. I recommend publication, but would like to have a few simple questions answered first.

Their MD methodology wasn't 100% clear to me outside of the basics. They look to determine material stability with MD, but how did they come up with 1% density change rule? Is this standard for MOF NPT calculations? For clarity, they should do a light expansion (e.g., perhaps a paragraph) on the MD portion. Why is this specific route and their metrics used? Are they standard for MOFs? This might be obvious to some readers, but not necessarily to all.

Response: Thanks, we appreciate this recommendation. We now include two new subsections in the Results section where we describe the suite of MD and GCMC simulations we used to identify high performing AI-generated MOFs: *Structural validation of AI-generated MOFs with molecular dynamics simulations* and *Property validation of AI-generated MOFs with grand canonical Monte Carlo (GCMC) simulations*. We also provide details of these simulations in the Methods section in the subsections *Structure Validation of the Assembled MOFs* and *Grand Canonical Monte Carlo (GCMC) Simulations*.

We agree that our previous metric of 1% change in all lattice parameters of the MOFs was somewhat arbitrary, and perhaps too stringent. Based on the analysis of Coupry et al. (J. Chem.

Theory Comput. 2016, 12, 10, 5215–5225), indicating that using the UFF4MOF force field 76.5% of cell parameters are within 5% of the reference value, and 95% within 10%, in the revised manuscript, we have relaxed this metric to 5% change in all lattice parameters, i.e., vector lengths and angles.

Several related notes that I hope they can discuss. They clearly set a target for accelerated discovery of materials with ML, but the amount of computational chemistry calculations they have to do after the ML is obscene. The ML generated 6000 structures to study via computational chemistry, more than 98% of the structures must have been absurd to fully disintegrate in MD. The MD found less than 1% of the suggested structures stable (18 out of 6000), and only 102 out of 6000 structures (less than 2%) that were even remotely reasonable.

Response: Thanks, we understand these concerns and have done the following to address them: 1) we added a new screening step, interatomic distance check to ensure the structural validity of AI-predicted MOFs. This approach boosts the fraction of chemically and structurally consistent MOFs; 2) we have included a detailed benchmark of the computational expense of our proposed approach in the Discussion section. At a glance, it only takes about 5 hours from assembling 120,000 new MOFs to downselecting 364 high-performing AI-generated MOFs. Out of these 364 MOFs, LAMMPS MD simulations indicate that 102 are stable and chemically consistent. We also used GCMC to estimate the CO₂ capacities of these 102 MOFs and present the top 6 MOFs in our revised manuscript.

How useful is AI that requires you afterwards to do MD simulations on 6000 structures, and after the MD you find that the VAST majority of those generated structures were useless and unphysical?

Response: We understand these concerns. Let's try to contextualize them: 1) from an experimentalist perspective, no MOF has been synthesized today that is adequate for carbon capture at industrial scale. This means that current approaches, used for at least a couple of decades, are not resulting in successful outcomes; 2) traditional large-scale MD simulations are compute-intensive and time-consuming. A single GCMC simulation takes longer than the process of creating 364 high-performing, chemically and structurally consistent AI-generated MOFs. The main contribution of AI is to reduce the search space by suggesting a fraction of potential molecules, which will be further tested by computational chemistry methods.

At the same time, the work we introduce here may be used as the foundation to create an automated workflow in which generative AI is coupled with DFT-quality simulations so as to increase the accuracy of AI-predicted MOFs through online learning methods. We have included these considerations in the manuscript.

These comments likely sound more negative than my intention, the final structures are quite interesting, and the procedure is logical and innovative. I don't think they need to solve these problems for publication. I do think though that they should at least more thoroughly discuss them, and better comment about the utility and scope of such an approach.

Response: We thank the reviewer for their candid assessment and suggestions! They helped us improve several components of our proposed framework, and to better describe our methods, their computational capabilities, and clearly state how these AI methods will be critical to transfer accelerated *in silico* modeling into synthesis pathways of novel MOFs.

REVIEWERS' COMMENTS:

Reviewer #1 (Remarks to the Author):

The authors already provided a very high-quality article describing generative AI approach to screen pcu-topology MOFs for carbon capture applications during the last revision stage. I am very pleased to see the huge efforts they put in improving their approach based on the reviewers' suggestions. In particular, the authors proved that their method is chemically and physically consistent with prior works in the field. The effect of catenation is much better demonstrated using their generative approach by comparing structurally similar MOFs with different levels of catenation. Furthermore, the best materials they found have been much better validated using standard more costly GCMC calculations.

For all these reasons, I recommend publication.

I hope the authors will make an experimental follow-up study on the novel (yet to be proven) best MOF materials from this screening to further prove the relevance of their approach.

Reviewer #2 (Remarks to the Author):

The authors have sufficiently responded to my requests. The manuscript looks to be in good form now. I recommend publication.